# Small scale characterization of vine plant root water uptake via 3D electrical resistivity tomography and Mise-à-la-Masse method.

Benjamin Mary[1], Luca Peruzzo[2,3], Jacopo Boaga[1], Myriam Schmutz[3], Yuxin Wu[4], Susan S. Hubbard[4], Giorgio Cassiani[1]

[1]Dipartimento di Geoscienze, Università degli Studi di Padova, Via G. Gradenigo, 6–35131 Padova, Italy
[2]GO-Energy, Geosciences Division at Lawrence Berkeley National Laboratory, Building 74, Calvin Road, Berkeley, CA, USA.
[3]EA G&E 4592, Bordeaux INP, University Bordeaux Montaigne, Pessac, France
[4]Earth and Environmental Sciences, Lawrence Berkeley National Laboratory, Berkeley, California 94720, USA.

*Correspondence to*: Benjamin Mary (benjamin.mary@unipd.it)

**Abstract.** The investigation of plant roots is inherently difficult and often neglected. Being out of sight, roots are often out of mind. Still, roots play a key role in the exchange of mass and energy between soil and the atmosphere, let alone the many practical applications in agriculture. In this paper, we propose a method for roots imaging based on the joint use of two electrical non-invasive methods, Electrical Resistivity Tomography (ERT) and Mise-a-la-Masse (MALM). The approach is based on the key assumption that the plant root system acts as an electrically conductive body, so that injecting electrical current in the plant stem will ultimately result in the injection of current in the subsoil through the root system, and particularly through the root terminations via hair roots. Evidence from field data, showing that voltage distribution is very different whether current is injected in the tree stem or in the ground, strongly supports this hypothesis. The proposed procedure involves a stepwise inversion of both ERT and MALM data that ultimately leads to the identification of electrical resistivity distribution, and of the current-injection root distribution in the three-dimensional soil space. This, in turn, is a proxy to the active (hair) root density in the ground. We tested the proposed procedure on synthetic data and, more importantly, on field data collected in vineyard, where the estimated depth of the root zone proved to be in agreement with literature on similar crops. The proposed non-invasive approach is a step forward towards a better quantification of roots structure and functioning.

# 1 Introduction

## 1.1 Problem statement

Soil root systems play a pivotal role in the many soil hydrological functions. Soil-plant interactions are complex, time-dependent, scale-dependent, species-dependent, and spatially heterogeneous. Special attention shall be paid to plant roots. It is therefore important to have techniques allowing to assess root system properties at the appropriate support scale.

## 1.2 Non-invasive measurements and electrical properties of root system

Non-invasive methods can provide spatially extensive, high-resolution information that, supported by traditional local data, help complete the complex picture of subsoil structure and dynamics. Among non-invasive methods, Grote et al. (2010) discussed the use of GPR for water estimation in a vineyard. However, depth of investigation of GPR is often limited by the soil type and is difficult to apply in clayed soil, and resolution is constrained by available wavelengths. A more developed approach is Electrical Tomography Imaging (ERI) - also called Electrical Resistivity Tomography (ERT) – that can be particularly informative regarding soil water content. In most soil types, electrical resistivity (ER) can be described as a function of porosity, saturation of electrolyte, its pH and mineralization within the pores (Archie, 1942), clay content (and generalized Archie's laws) and temperature (e.g. Campbell et al., 1948). Water content could be derived from the measured ER using pedotransfer functions such as the well-known Archie's law or other approaches (e.g. Rhoades et al., 1976; Waxman and Smits, 1968). Since absolute soil moisture content is of limited interest for researchers and professionals which are focused on soil water availability for the plant, several studies relate the application of the variation of ERT or the Fraction of electrical resistivity variation (FERV) introduced by Brillante, et al. (2016) as a predictor of Fraction of Transpirable Soil Water (FTSW) and related variables.

Amato et al. (2009) tested the capability of 3-D ERT to quantify root biomass on herbaceous plants using resistivity root correlation/calibration. Electrical methods have been also used to identify Root Water Uptake (RWU - e.g. Cassiani et al., 2012; Garré et al., 2011; Michot et al., 2003; Srayeddin and Doussan, 2009) and demonstrated the match between soil water content variations and temporal changes in electrical resistivity. Cassiani et al. 2016 monitored the electrical resistivity in an apple orchard under external forcing conditions (irrigation and plant driven evaporation) and showed that the increase of resistivity is located in the subsoil region where active roots are present. Electrical and electromagnetic methods have been also used to identify Root Water Uptake (e.g. Cassiani et al., 2015). Werban et al. (2008) performed an interesting ERT study on lupine roots and showed that rooted soil differs from bare soil in terms of pedo-physical model. Several studies related to soil-root systems have shown that the measured root mass density statistically correlates with the electrical conductivity (EC) data obtained from ERT (Amato et al. 2009). Nevertheless, in some cases, the ranges of electrical resistivity of soil and roots overlap. The amplitude of contrasts varies according to the soil resistivity and tree species (Zanetti et al. 2011; Mary et al. 2016), to the water content and the decay state of the wood itself (Martin 2012) and to variations in soil water content (Garré et al. 2011; Beff et al. 2013; Cassiani et al. 2015; Mary et al. 2016). The problem would complicate further the correlation with root mass considering heterogeneous soil properties and moisture, and the electrical anisotropy caused by the roots system i.e. the root connectivity and root structure as further described in Rao et al. (2018).

Recent studies have shown a correlation between bulk electrical resistivity and root mass density, but an understanding of the contribution of the segments of the root system (by its own properties, with no interaction with soil) to that bulk signal is limited to only few studies describing wood electrical properties. Gora et al. (2015) reviewed the literature describing electrical properties of stems noting large differences between trees and vines plants resistivity values (200% higher for trees), suggesting that there is a phylogenetic basis for variation of the ER that reflects influence of anatomy and physiology. Observations from stems are directly transposable to roots. The range of electrical resistivity of roots depends on their nature. Typically, coarse roots, because of the heartwood and the isolative layer of bark, are considerably more resistive than fine roots (Hagrey, 2007).

Electrical resistivity is also linked to the physiological state of the roots. Depending on the season, roots carry electrical charges as sap composition is variable and sap flows vary in intensity and direction. Wood composition and physical properties also change with root decay which implies a variation of electrical properties (Martin, 2012; Martin and Günther, 2013; Weller et al., 2006). Very recently, Rao et al. (2018) produced an interesting study aiming at understanding individual root segment on bulk electrical conductivity, incorporating the impact of root in the pedophysical relations to better infer the real soil water content.

Finally, root water uptake and the release of different exudates by fine roots change soil water content and resistivity at several temporal scales (York et al,. 2016): on a daily basis (night vs day, sunny vs cloudy days) and seasonally (growth period vs winter or drought season). In conclusion, roots might have a considerable impact on ERT signals, but this may not be directly measurable: ERT thus is an indirect determination of roots presence.

Other bio-electrical phenomena can contribute to a more effective characterization of root properties. Plant water uptake generates a water circulation and a mineral segregation at the soil-roots interface, thus inducing an ionic concentration gradient which generates an electrical potential of few mV. This can be measured in terms of a passive distribution of voltage in the soil (Boleve, 2009). Gora et al. (2017), provide a framework for studying the ecological effects of lightning in the context of electrical properties of tress. Also, Gibert et al. (2006) and Le Mouël et al. (2010) measured natural variations of electrical potential using electrodes into the stem. They respectively measured 25mV due to daily variations of sap flow and 10 to 50mV caused by the flow of thunderstorms which would produce soil charges and give rise to a current circulating through the roots and the tree stem. In theory, by analyzing the voltage distribution in the soil, it is possible to find the characteristics of the sources (depth and extension) causing the voltage anomaly (Saracco et al., 2004). In practice, these sources are too low in generally noisy environments. But one may think about taking advantage of these results to build an active method producing current flow and potential distribution into the soil.

### 1.3 The Mise-à-la-masse method applied to plant root systems.

In this study, we aim at investigating the feasibility of the "Mise-à-la-masse" (MALM) method in the context of plant roots mapping. MALM is an electrical resistivity method originally developed to delineate conductive ore bodies for mining exploration purposes (e.g. Schlumberger, 1920; Parasnis, 1967). An electrical current is injected in a conductive body and the resulting voltage values are measured at the ground surface or in boreholes; the shape of equipotential contour lines is informative on the extent and orientation of the conductive body in the subsoil.

In the plant stem and roots, electrical current is transmitted through active electrical layers, in the xylem and phloem (on either side of the cambium), where sap flow processes take place. Our main assumption is to consider that thanks to the quasi-infinite fine root connections and their mycorrhizal at the interface between roots and soil, current tends to run out uniformly from the roots to the soil. In the context of MALM applications, the tree root system can thus be viewed as the conductive body to be imaged, with some important caveats: current may be carried within the roots, but are likely to be released into the soil only at the points where fine roots emerge from the woody root structure. This would be of major interest to measure for plant science community as fine roots are the active ones. Note that this is not necessarily proven for non woody plant species. For instance, Anderson and Higinbotham (1976) showed that maize roots have significantly lower electrical resistance in the radial than in the axial direction (thus being anisotropic), thus allowing current to exit laterally from the entire root length.

In practice, for MALM applied to roots prospection, the current is to be injected directly into the tree stem with one electrode, while the other current electrode is placed in the soil at some distance from the tree. Voltage is measured at the soil surface and in boreholes with respect to a second, remote reference voltage electrode.

It must be noted that soil and roots conductivity depends, among other parameters, on seasonal variations, water content or even salinity of the soil making the interpretation potentially complex. On the other side, the sensitivity of MALM to water content make the monitoring of plant water uptake occurring near roots possible and strengthens interpretation of their location.

Some knowledge gaps exist concerning root electrical properties. However, several theories have been proposed in the scientific literature in this respect, all confirming that each root may act as a current source in the MALM configuration above:

- Dalton (1995) analyzed the root-soil circuit and proposed a conceptual model with an electrical analog composed of resistance R and capacitance C (the ability of a system to store an electric charge). In that model, the internal fluid (xylem and phloem) of the plant roots constitute a duct of low resistance which is separated from a low resistance external medium (soil) by insulating root membranes (Ozier-Lafontaine et al., 2005). These membranes, in addition to being insulating, accumulate charges on the surface. Observation from Dalton and subsequent theories are fully consistent with the use of MALM for plants, even though the capacitance part is not exploited in these measurements. A benchmarking of the experimental approaches supporting the subsequent theories is proposed in Postic et al., (2016).

- The second theory is based on the notion of absorbing root surface and developed in the studies of Aubrecht et al. (2006) and Cermak et al. (2006). These studies indicate that if a plant growing on soil is connected to a simple serial electric circuit, then current flowing through this circuit from the external source enters the plant entirely through the absorption zones (or vice versa). Electric current can also flow through impermeable walls of other cells, but with a negligible density.

- A third theory is based on roots polarization of biomatter as a proxy of root current pathway. As previously mentioned, root systems are commonly modeled using electrical circuit composed of resistance R and capacitance C within the Dalton (1995) and similar refined models (Aubrecht et al., 2006; Cermak et al., 2006). This means that the conduction of the current through the root system depends on current characteristics. For alternating currents (AC), resistivity is complex number for polarizing medium having resistance and capacitance parts and is therefore dependent on frequency, and a phase shift occurs. This shift is dependent on some specific plant parameters, and its assessment could also contribute to better discriminate roots and soil current conduction. Mary et al. (2017) considered polarization from soil to root tissues, as well as the polarization processes along and around roots to explain the phase shift observed for different soil water content. Weigand et al. (2017) demonstrate that multi-frequency electrical impedance tomography is capable of imaging root systems extent as well as to monitor changes associated with root physiological processes.

Given the review of current knowledge on electrical properties of roots, in this paper we hypothesize that the mise-à-la-masse method can be a viable tool to locate active roots under in situ conditions. The paper has the following aims:

(1) define a viable field protocol that uses jointly MALM and ERT to map active tree vine roots;

(2) propose and analyze algorithms capable of identifying the location of active roots;

(3) test the algorithms above against real data from a French vineyard.

A discussion of the results will be provided in the light of biological assumptions.

## 2 Materials and methods

### 2.1 Site description

The field study was carried out in a vinery of the Chateau La Louvière appellation controlée, located in Pessac-Leognan (Figure 1a) near Bordeaux (Gironde, France). The climate of the region is Oceanic with average annual air temperature of 13.7°C and total annual rainfall of 811 mm (André et al., 2012). According to the meteorological station nearby the experimental plot (200m), the study period was wet consecutive to rainfall with an air temperature of 11 °C. The topography of the plot is mostly convex with an average slope of about 10% but less than this value at the location of the experimental plot, thus inducing small surface water runoff.

### 2.1.1 Soil characteristics and prior knowledge on root systems

Despite heterogeneities of soil types composing the vineyards (André et al., 2012), the plot location is all located in a similar soil system (Baize, 1998). Moreover, the organization of the soil sequence and roots density was investigated with observation trenches. The closest one to the experimental plot shows an organization, with a first sandy horizon (0 - 40 cm depth), porous and soft. Rooting depth has been qualitatively observed on a bare soil at the emplacement of uprooted vine plants and can been only seen as ancillary information's. In this horizon, all root sizes with a rather horizontal and oblique orientation were observed. The second layer (40 - 105 cm depth) is identical to the top layer in terms of soil composition, but contains less roots. A third layer (deeper than 105 cm) is relatively similar to the previous ones with only very few fine roots. From 125 to 175cm depth the soil type changes to sandy-clayey. The described geology, morphology and microclimate of the regional context defines the so-called terroir de grave of this vineyard, where vine plant species have been planted. For this study, we selected an apparently healthy plant. Considering the soil composition, the vine water supply is facilitated thanks to the possible capillary rise from the sandy-clayey horizon which retains sufficient water for vine use and generally contain sufficient nutrients for vine growth. Grapevine plants are planted with a distance of 1m between plants and 1.5m between rows. The vineyard is non irrigated. Considering also the selected plant and the slight slope of the vineyard, it might be reasonable to foresee a top layer rooting with an asymmetric development (gravitropism).

### 2.2 3-D scheme of ERT/MALM set-up acquisition and processing

The 3D ERT setup was originally developed by Boaga et al. (2013) and subsequently improved and adapted at different sites to obtain successful results regarding soil-plant interactions, e.g. in salt marsh environments (Boaga et al., 2014) and in apple and orange orchards (Cassiani et al., 2015; Cassiani et al., 2016; Consoli et al., 2017; Vanella et al., 2018). The apparatus was adapted again and applied for the first time in a vineyard for this study. Figure 2 shows the geometry of the electrode system: 4 micro-boreholes define a rectangular domain, 1m along the vineyard direction (y), and 1.2m in the perpendicular direction. Each borehole houses 12 electrodes with 0.1 m vertical spacing. In addition, 24 surface electrodes define a regular grid. Such disposition allowed us to conduct high-resolution measurements around the selected vine plant (

Figure 2). Field measurements were conducted in March 2017, using a ten-channel resistivity meter (Syscal Pro 72 Switch, IRIS Instruments). For the 3-D scheme of ERT, a complete skip-two dipole-dipole scheme was adopted and produced some 5000 measurements, including reciprocals measurements used to estimate and reject bad data quality (Binley et al., 1995; Daily et al., 2004). A pulse duration of 250 ms for each measurement cycle and a target of 50 mV for potential readings, were set as criteria for the current injection. The R3t code (Binley, 2013) was used for data inversion.

### 2.3 MALM acquisition, modeling, and processing

### 2.3.1 MALM acquisition and forward modeling

The MALM acquisition used the same electrode arrangement as described above, with only a couple of necessary changes: the two remote electrodes for current (B) and potential (M) – see Fig. 2 – were located at a large ("infinite") distance, more than 20 times the maximum distance between the other electrodes, as suggested by Robain et al. (1999). An additional electrode was placed near the stem (

Figure 2). Two different datasets were acquired depending on the position of the current injection electrode A, as described in the workflow in Figure 3: (i) the first case, was a real MALM acquisition where the injection electrode A was planted into the apparent conductor (i.e. the plant stem); (ii) the second case is a reference (or false MALM) case, with the injection electrode A was planted in the soil very close to the stem. A comparison between the two acquisitions is expected to show the effect of the plant as a current conveyer. All surface and borehole remaining electrodes (69) are used to measure voltage. Compare to Pole-Dipole used for capacitive measurements with two electrodes implanted into the stem (Aubrecht et al. 2006 and Cermak

et al. 2006), there is no additive stem resistance to consider and this fact is particularly important for the data interpretation. However, good contact of the electrode with the stem must be ensured for the "true" MALM acquisition: the current electrode was planted about 1 cm into the 5 cm wide stem, thus reaching the cambium layer, and ensuring a stable contact resistance of about 10 kΩ. Reciprocals measurements were also acquired in MALM (Appendix, Fig. A.1). In order to compare voltage data against simulations (see below), values of the potential measured on the surface of the ground and in depth with boreholes were systematically normalized by the amplitude of the injected current.

Synthetic MALM data were produced (in forward mode) using the R3t code (Binley, 2013) and the same unstructured tetrahedral mesh used for ERT inversion. The quality of the meshing was checked by the comparison between a uniform 100 Ωm forward modeling with the corresponding analytic solution (Appendix, Fig. A.2). Inverted resistivity from 3D ERT acquisition was considered as a resistivity distribution needed for the MALM forward modeling.

### 2.3.2 Processing and interpretation using objective functions

In order to interpret the results of the MALM acquisitions, a quantitative inversion of the voltage data is needed. This requires (a) the forward simulation of voltage values given a certain current source distribution in the soil (equivalent to the locations were current reaches the soil emerging from the roots); (b) the minimization of an objective function that defines the discrepancy between measured and predicted voltages, where the minimization variable is the location of the electrically active roots. Steps (a) and (b) are equivalent to inverting the data for the current source distribution in the soil, which in our conceptualization also represents the distribution of active (fine) roots in the system.

In the following, two different objective functions are introduced. First, assuming that a unique current punctual source is sufficient to fit the measured data, the following objective function is to be minimized:

$$F1\left(D_m, D_{f,i}\right) = \left\| D_m - D_{f,i} \right\|_2 \tag{1}$$

where $D_m$ are the measured voltage (V) and $D_{f,i}$ the forward voltage data for one source positions (i-th node of the mesh). The F1 function can help guide the search for the region where the presence of active source is most likely to concentrate, but of course the use of F1 alone does not represent a realistic distribution of sources in the MALM inversion.

A more realistic objective function, that takes into account the presence of distributed sources, has also been introduced:

$$F2\left(\alpha_i, D_m, D_{f,i}\right) = \left\| D_m - \sum_{i=1}^{Ns} \alpha_i \times D_f \right\|_2 \tag{2}$$

where $0 < \alpha_i < 1$ expresses the contribution of source $i$, with the constraint $\sum_{i=1}^{Ns} \alpha_i = 1$ , with Ns the total number of current sources that ensures that the electrical charge (and thus the electrical current) is conserved. The number of current sources to invert for (Ns) is primarily dictated by the desired input mesh quality (Fig.A2c). This is determined by the required computational time. For this small scale prospection, we adopted a mesh composed of 23700 nodes (including remotes electrodes, e.g. Fig.A2a). The inversion region was limited to 3618 nodes (Fig.A2b). Furthermore, as shown in Fig. 3, the strategy is to use the F1 and F2 optimizations sequentially. In order to guide the physically sound F2 inversion, initial values of $\alpha_{io} = [\alpha_{1o}, \alpha_{2o}, ..., \alpha_{Nso}]$ were set using normalized F1 values (between 0 and 1). This is equivalent to applying a regularization based upon the initial F1 search upon the F2 optimization. A global optimization using a Constrained Nonlinear Optimization Algorithms (*fmincon* solver using a gradient-based method associated with the Sequential quadratic programming (SQP) optimization algorithm) method implemented in MATLAB® (R2016b) software was then used to minimize F2.

### 2.3.3 Testing of the inversion procedure: a synthetic data example

In this synthetic example, we used the same configuration, mesh and protocols as for the real case (see Sect. 2.2). Figure 4 shows the initial model with the location of a cubic resistive anomaly (500 Ω.m) embedded in a 100 Ω.m background. The

anomaly is slightly shifted compare to the acquisition domain. A dipole-dipole skip 2 protocol was adopted for ERT acquisition (Fig3, step 0). The same mesh was used for ERT and MALM simulations.

The resistivity distribution obtained from ERT was used, as necessary in real cases, as the background resistivity through which the current, induced by the MALM experiment, flows. In this synthetic example, the MALM datasets (see Fig. 3 – ERT Model) are obtained hypothesizing current source locations (at the FE mesh nodes) within the given theoretical "root" zone – the current intensity is assumed the same at all nodes. Figure 4 shows the distribution of the voltage, solution of the sum of the contribution, measured with surface (Fig. 4c) and boreholes electrodes (Fig. 4d). Results from F1 minimization allowed for a preliminary selection of the region where individual sources should be considered for weight distribution in F2 minimization. The minimum number of sources was selected according to the evolution of the curve of sorted misfit F1 (the same procedure applied to the real data, see Sect. 3): any increase in the number of candidate source locations does not decrease significantly the F1 value. In this synthetic case, a minimum misfit F1 reaches a value of 17%, and the corresponding contours of the F1 objective function (Fig. 5a) indicate the volume of the "true" anomaly. This step results in the selection of probable sources defining a preferential search space area for the subsequent F2 minimization.

Sources weight results inferred from F2 minimization (distributed weighted sources assumption) were then sum to compute an inverted model. Figure 5 shows the solution, the inverted model for surface and boreholes electrodes for the synthetic case. The asymmetric nature of the solution is clearly visible from both surface and borehole electrodes.

## 3 Experimental results

### 3.1 Field 3d ERT measured data

Figure 6 shows the solution of the inversion from the 3D ERT data acquisition. The pulse duration was 250 ms per measurement cycle, and the target voltage was 50 mV for the current injection. The result of a measure corresponds to the mean of between 3 and 6 stacks with a relative difference between two stacks of 5% on the resistivity term. Contact resistances were good during the acquisition: by accepting a threshold equal to 5% for reciprocity error, only 12% of the measurements were rejected. Electrical resistivity ranges from 100 to 250 $\Omega$m with significant lateral and vertical spatial variations (Figure 6). Soil texture is expected rather homogeneous with depth, except at the very top where the soil tillage can induce also electrical resistivity changes. A profile taken at 0.2 m depth (Fig. 6a) shows two distinct peaks of resistivity; the first peak corresponding to the highest value of ER (250 $\Omega$.m) is located at y=0.78m, close to the plant stem position but with some slight shift. In the 3D visualization (Fig. 6c) the high resistivity peak corresponds to an extended anomaly around the plant.

When considering the electrical resistivity profile with depth below the stem (Figure 6b), a maximum region between 0.2 m and 0.4 m depth is clearly visible. A horizontal profile at 0.4 m depth (Fig. 6a) confirms a maximum around y=0.7 m, not far from the stem location. At larger depths no noteworthy features are apparent since neither soil tillage nor plant roots seem to act on the electrical resistivity of the soil.

### 3.2 MALM results

As discussed above, we acquired direct and reciprocal measurements also for the MALM data. A comparison between direct and reciprocal resistances allows, in ERT, to quantify the data quality, remove outliers and define the error level to be adopted in the data inversion procedure. However, the reciprocity theorem holds only in case of linearity (Parasnis, 1988). In the MALM case at hand, linearity may be violated when current is injected in the tree stem (by accepting a threshold equal to 5% for reciprocity error, only 10% of the measurements were rejected for the stem injection while 7% for the soil injection). And indeed the differences between direct and reciprocal data (Fig. A.1 in the Appendix) seem to be systematic and linked to the region around the stem. In the following we will refer to the MALM results obtained by injecting current in the stem. Figure 7 shows a comparison between normalized voltage data obtained by injecting in the stem and in the soil. At a very first glance,

the spatial distributions of the voltage caused by stem and soil injection results appear very similar, with the striking exception of the voltage absolute values, with the stem injection leading to much lower normalized voltage values (maximum is 200 V/A versus 500 V/A for the soil injection) especially close to the stem. This is an indication that current is indeed not injected at the ground surface, but emerges at some point(s) below ground. Note that also the gradients along the ground surface are much

steeper for the soil injection than for the stem injection, confirming the hypothesis just presented.

Figure 7 also shows borehole results, which appear more complex to interpret in terms of actual current distribution. Normalized voltages range between roughly 20 V/A at 0.1 m depth to nearly zero at 1.3 m depth. For both stem and soil injections, the voltage decreases regularly from 0.6 m to 1.3 m. Slight differences in the decay slope and between boreholes are only visible for the shallow region (0-0.6 m depth). In particular, in presence of stem injection, the voltage is nearly constant

from 0 to 0.3 m depth, while for soil injection the slope is slightly larger. This pattern is observed in each borehole. Borehole 4 shows some irregular behavior (one electrode is abnormally low, possibly because of bad contact with the soil. On average, voltages resulting from soil injection are higher than from stem injection.

### 3.3 Inversion of MALM field data: punctual source search (F1 function)

Figure 8 shows the spatial distribution of the F1 function, where the spatial dependence is implicitly accounted for by the index

$i$ ($i$-th node in the mesh) in Eq. (1). Each individual source was forwarded to produce a tentative normalized voltage at electrodes also as a function of the resistivity distribution reconstructed by ERT inversion of field data. Obviously, figure 8 shows that none of the single source positions is capable of fitting all data perfectly – the misfit range reproduced by F1 values in Figure 8 is between 10% and 50%. Nevertheless, the fit is not too low, and the F1 spatial distribution is a clear indication of the regions where distributed sources shall be placed to reproduce field data. For both injection schemes, in stem and soil,

F1 values decrease with depth, but with different rates. In the case of injection in the soil, the source locations with a 20% misfit are very close to the ground surface (within 0.05 m depth). In the case of stem current injection, the same misfit level extends to 0.3 m depth (Fig. 8c).

### 3.4 Inversion of MALM field data: distributed sources (F2 function)

Considering a single punctual source is, of course, a very rough approach in trying to identify the distribution of current sources

that generate the observed MALM voltage distributions. Thus we used the results of the section above only as a first approach to guide the identification of distributed current sources. The objective function in Eq. (2) - named F2 – was used for inversion of sources during stem current injection. Function F2 reflects the L2 norm of the differences between the measured data and the sum of the sources weighted by a coefficient α that is accounting for the fraction of total current pertaining to that source. The vector of α values is the target of the inversion, while the locations of candidate sources is defined by the nodes of the

Finite Element mesh used for forward modelling. Given the very large number of nodes, most of which are located in regions that are very unlikely to host active roots, and thus MALM current sources, we constrained the candidate locations on the basis of the results of the F1 inversion (see section above): only locations that would contribute in a substantial manner to reducing the F1 misfit (to 17%) are used as candidate locations in the F2 minimization – about 200 locations were used (see Fig. 8c). The corresponding values of optimized F1 are used, after their sum is normalized to 1, as initial guesses for α values to start

the inversion. Individual α are allowed to vary in the 1e-4 to 0.1 range. Current conservation was respected since the sum of weight was equal to 1 at the end of the inversion iterations.

The result of the F2 minimization is shown in Figure 9, where it is apparent how the region where distributed current sources are located is no deeper than about 0.3 m, and has a lateral extent between 0.5 and 0.9 m. This is likely to be the extent of the plant active roots.

## 4 Discussion

This study shows how the joint use of ERT and MALM can help the characterization of a plant root system. However, while we show how substantial progresses can be made, it is apparent that a number of tricky details must be considered and further developments are needed. Our work clearly shows that the MALM method can provide key information concerning the root system spatial distribution of woody species (with the latter discussed uncertainties). This is apparent from the simple comparison of (normalized) voltage distribution as produced by current injection in the soil and in the plant stem (e.g. Fig. 7). However, the differences in normalized voltage between stem and soil current injection, even though apparent, are not such as to evidently point towards a self-evident distribution of current sources to be associated in an obvious manner to an active root distribution. Thus we must go beyond a simple qualitative approach.

Modelling has been used recently to bridge the gap between simple voltage measurements (MALM) and complex three-dimensional inverse modelling (ERT). The gap is caused essentially by the relative scarcity of data inherently linked to the MALM acquisition as compared to the wealth of data generally acquired in ERT (and especially in 3D ERT) acquisitions. Recent examples are given e.g. by De Carlo et al. (2013) and Perri et al. (2018). In all cases, forward modelling of MALM is used to compare simulated and measured data, given certain assumptions concerning, usually, the distribution of electrical resistivity in the subsurface – since injected current locations are known. In the present case, we exploited modelling in a different manner, taking full advantage of the joint availability of MALM and ERT data on the same configuration. As in other MALM studies, the modelling exercise is used to test some underlying assumption: in this case, we assume that injecting current in the plant stem causes a distribution of electrical current sources in the ground that corresponds to the locations of active roots, i.e. to the locations where roots are in contact with the ground also in terms of electrical conductance. The fact that this contact does not correspond to the place where the plant stem touches the ground is verified by the simple comparison between stem and soil injection – that produce different MALM voltage distributions. The modelling exercise is actually set up as an inversion process, as in our case we only aim at identifying current injection locations, as the electrical resistivity distribution is assumed to be known from the independently acquired 3D ERT results. In practice a double inversion is carried out: (1) ERT data are inverted to give the estimated electrical resistivity distribution; (2) assuming that the ERT-derived resistivity is correct, stem-injection MALM data are inverted only for the locations of current sources.

The procedure above is not free from uncertainty. In particular:

- The identification of current source locations is inherently an ill-posed problem, as the number of candidate locations is potentially very large and the current intensity for each injection point is of course unknown. Given that the MALM normalized voltage is only measured at a very limited number of electrodes, we cannot expect that a unique solution is possible. However, the space of possible solutions can be constrained and volumes of likely current injections can be identified, as we demonstrate both in the synthetic and real cases above.
- The electrical resistivity distribution in the ground has a strong impact on measured MALM voltages. In this respect we can only trust the effectiveness of ERT in identifying this distribution, at least within the precision needed for its use in MALM sources inversion.

The two points above have the consequence that the overall minimization of objective functions F1 e.g. Eq. (1) and F2 e.g. Eq. (2) cannot lead to very small misfit values, especially if the possible distribution of sources for F2 is constrained a priori by the F1 distribution. We accept that the resulting misfit is a measure of the limitations inherent in the assumptions made.

The main assumption that is made is that the root system acts as a preferential electrical pathway, with current flowing inside the conductive parts of the roots (xylem and phloem), and thus preventing the release of the current from roots to soil across the roots woody outer bark. The current is ultimately discharged to the soil by the multitude of thin/hair roots. In practice, more research should be conducted in order to establish whether the current is going through the entire roots system and how the vast number of hair roots contribute to the release of current. Water acquisition and by prolongation electrical current pathway is thought to be limited to the surface located close to root tips. At least two other phenomena may contribute to

current release higher than expected. Firstly, Cuneo et al., (2018) show that although woody portions of roots act as an electrical barrier (also to microbial degradation), exchanges may occur during water uptake can occur through (in order to facilitates localized embolism repair in grapevine). Secondly, as discussed also in the introduction, some roots show anisotropic electrical conductivity, allowing current to flow radially more easily than longitudinally (Anderson and Higinbotham, 1976). In this

case, our proposed MALM approach would need to be modified in the interpretation stage. Note that roots are generally electrically anisotropic at the microscopic scale (few cm) and also macroscopically the root architecture and soil water uptake pattern can induce anisotropy. Using MALM to study anisotropy of root structures can indeed be a separate, very promising, area of research. Note that the presence of electrical signals, such as action potentials (AP), in plant cells suggested that ion channels may transmit information over long distances (Pyatygin et al., 2008).

The results of our field study, albeit within the uncertainties just described, identify the presence of current sources, and thus likely the roots system, within the top 30-40 cm depth. This is not totally unexpected, even though we observe a slightly shallower range than usually reported in the literature dealing with wine roots system (Stevens and Douglas, 1994, Gerós et al., 2015). Moreover, roots with a diameter ranging from 0.5 to 2 mm, which have water and nutrient foraging and uptake functions (Herralde et al., 2010), represented the majority of the total, in mean more than 80% in most studies of grapevine

cultivars (Swanepoel and Southey, 1989; Morlat and Jacquet, 2003; Nagarajah, 1987). This is in agreement with our assumption that a vast number of small current sources correspond to the roots distribution. Finally, it is well known that fine, medium and woody root are not adequacy distributed with depth and the number and the diameters of the roots show a drastic decline with depth (Morano and Kliewer, 1994; Morlat and Jacquet, 2003, Tomasi et al., 2015). Our results are in clear agreement with this pattern that is mirrored by the decrease of $\alpha$ with depth. Although the rooting depth obtained in our study

reach approximately 0.3 - 0.4 m below ground, there are probably still roots growing below this depth. Their contribution to the MALM data is too low to be detected above the thresholds we applied for inversion, indicating a very small roots density and the resolution limit of the MALM method. From this observation, one can consider a correction during the inversion process using a depth weighting matrix. Nevertheless, in that study mathematical assumption should not be too much incriminated. If the rooting depth increase, the acquisition may take advantage of the boreholes preventing losing too much

resolution. We previously discussed potential sources of errors as we lack a convincing ground truth for individual root segments due to the inadequacy of existing direct investigation methods. Indeed, excavation (e.g. via air spade), although very performant for container-grown plants, are only a good way of showing the large roots, but much less their functioning in the field. Showing the woody roots is, for the most part, providing information on the structural support of the tree while RWU is controlled by fine structures that are in connection with the woody roots, but do not necessarily coincide totally with them.

Already Dittmer (1937) reports that living root hairs (Secale cereal L.) may be scattered over the entire surface of all the roots, nevertheless their relative number and length varied within the different root categories, and the smallest but most numerous were found in the quaternary division. Judd et al. (2015) reviewed the most frequently used field methods to measure or to analyze root systems and report that hair roots are destroyed during the field excavation using trench, window (Böhm, 1979), pinboards, and monoliths. Furthermore, these methods are static. As for the air spade, it has been widely used, but can even

damage the coarse roots (Stokes et al. 2002).  Existing less destructives methods such as auger core or (mini/meso)-rhizotron can show aberrant root growth along the walls or windows and requires a large number of samples or tubes (Taylor et al., 1990), and of course these methods are not applicable in the field.

A number of applications that would benefit from knowing the location and activity of roots may emerge from our proposed approach. Among others, the refinement of allometric root-shoot factors to study competition between plants, the improvement

of models for estimation of water available for plants (such as the FERV introduced by Brillante et al., 2016, as a predictor of FTSW), and the refinement of water balance modeling by assimilation of geophysical data (e.g. Manoli et al., 2015; Rossi et al., 2015).  On issue that has not been addressed in this study is how roots conduct electrical current depending on the plant physiological state. Seasonal variations would significantly affect the ions content and intensity of sap flow. During the

experiment in March the plant probably develops new roots and leaves (lateral shoot growth). The study period was wet consecutive to rainfall with an air temperature of 11°C. Conditions of the experiments were not optimized to fully highlight the root system. Limited water uptake was occurring during the experiment since the plant was not stressed. Sap flow was probably reduced, and so the resistivity of living plant tissues may have increased. Considering phenological phases of the plant may significantly improve the efficiency of the MALM approach we describe. A possible improvement would consist in using MALM to monitor an irrigation experiment or processes occurring after a rainfall event.

## 5 Conclusions

In this paper we present evidence showing how the joint use of MALM and ERT in a high-resolution, 3D configuration around a tree (in this case a vine) can provide very detailed information about the plant root system. The results are based upon the hypothesis that current injected in the tree stem is conveyed through the root system and released in the ground at the locations where hair roots are in electrical contact with the soil. This hypothesis is fully supported by existing scientific literature. In addition, our experiments show that the injection in the stem produces a very different voltage distribution than the injection directly in the soil at the base of the stem: this is solid evidence that the plant structure redistributes current in the soil, and this can only happen through the root system.

In order to produce quantitative results concerning the root system structure, we adopt a three-step inversion process:

(1) a 3D ERT inversion provides the spatial distribution of electrical resistivity as an indirect correlation of root biomass;

(2) a single-point MALM inversion, produces a 3D distribution of misfit values that is a measure of how likely is that a current source (read: a root) is present at that location;

(3) a multiple point MALM inversion, produces a 3D distribution of electrical current injection in the soil, that is the most likely proxy to the hair root distribution density in the soil.

While a number of pending issues remain to be discussed and developed in future work, the step forward is substantial and paves the way for the widespread use of electrical methods applied to study root-soil interactions. This, in turn, may lead to the successful pursuit of a number of possible practical and theoretical results. Among future developments, further work needs to be conducted to establish solid links between the proposed method and the plant physiological state. Modelling study with an explicit representation of root structure in the MALM forward modelling may be done as a follow up work to understand how the proposed approach can be made more robust.

## Acknowledgements

The authors wish to acknowledge support from the projects ERANET-MED WASA "Water Saving in Agriculture: Technological developments for the sustainable management of limited water resources in the Mediterranean area" and "Hydro-geophysical monitoring and modelling for the Earth's Critical Zone" (CPDA147114) funded by the University of Padua. In addition, the information, data, or work presented herein was funded in part by the Advanced Research Projects Agency-Energy (ARPA-E), U.S. Department of Energy, under Work Authorization Number 16/CJ000/04/08. The views and opinions of authors expressed herein do not necessarily state or reflect those of the United States Government or any agency thereof. . Luca Peruzzo and Myriam Schmutz gratefully acknowledge the financial support from IDEX (Initiative D'EXellence, France), the European regional development fund Interreg Sudoe - soil take care, num. SOE1/P4/F0023 - Sol Precaire.

# Appendix

## A.1 Reciprocals measurements

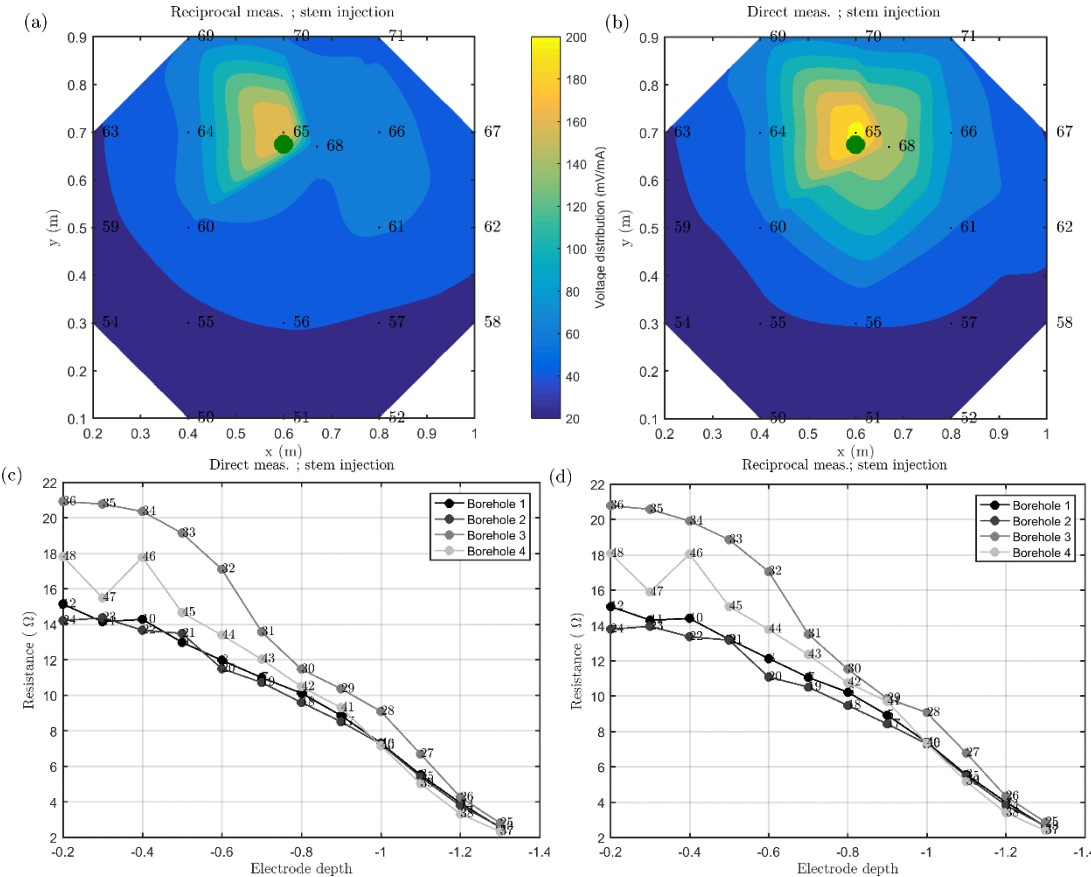

**Figure A.1: Spatial variations of the normalized voltage (I/U expressed in V/A) observed by surface electrodes (a and b, interpolated points) and boreholes electrodes (c and d) obtained during the MALM field measurements: direct measurements (current injected into the stem) are shown on the right, while reciprocals are shown on the left. The green dot shows the location of the plant stem (at x=0.65 m, y=0.67 m).**

## A.2 Mesh quality check

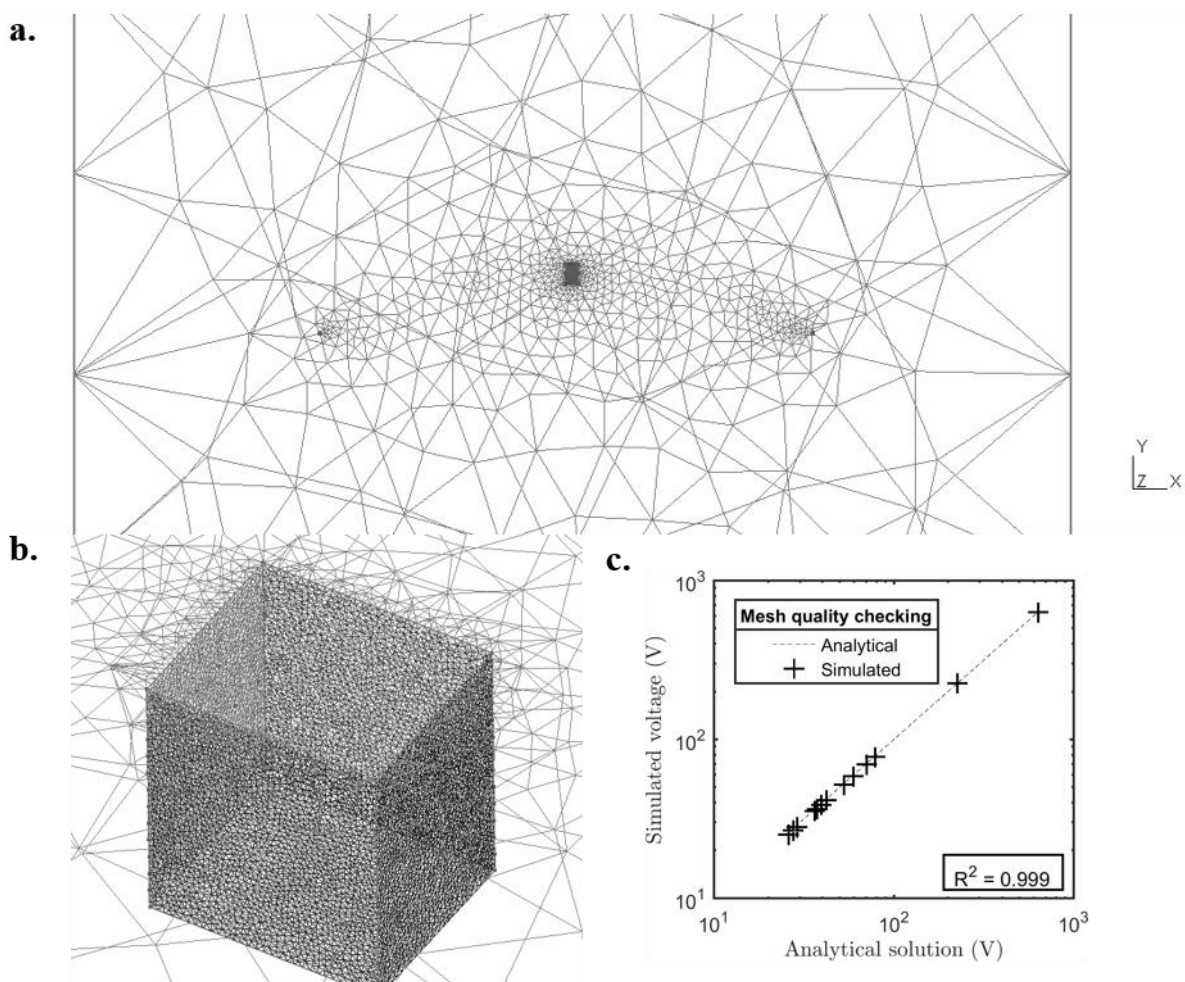

**Figure A.2: Plot of the finite element mesh used in this paper, showing: (a) position of remotes electrodes in the mesh and position of the stem; (b) zoom around the stem showing a mesh size approximately 5 times smaller than electrode spacing; (c) the plot showing the excellent correlation (R2=0.99) between numerical simulation results and analytic solution for a homogeneous model with resistivity equal to 100 Ωm.**

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

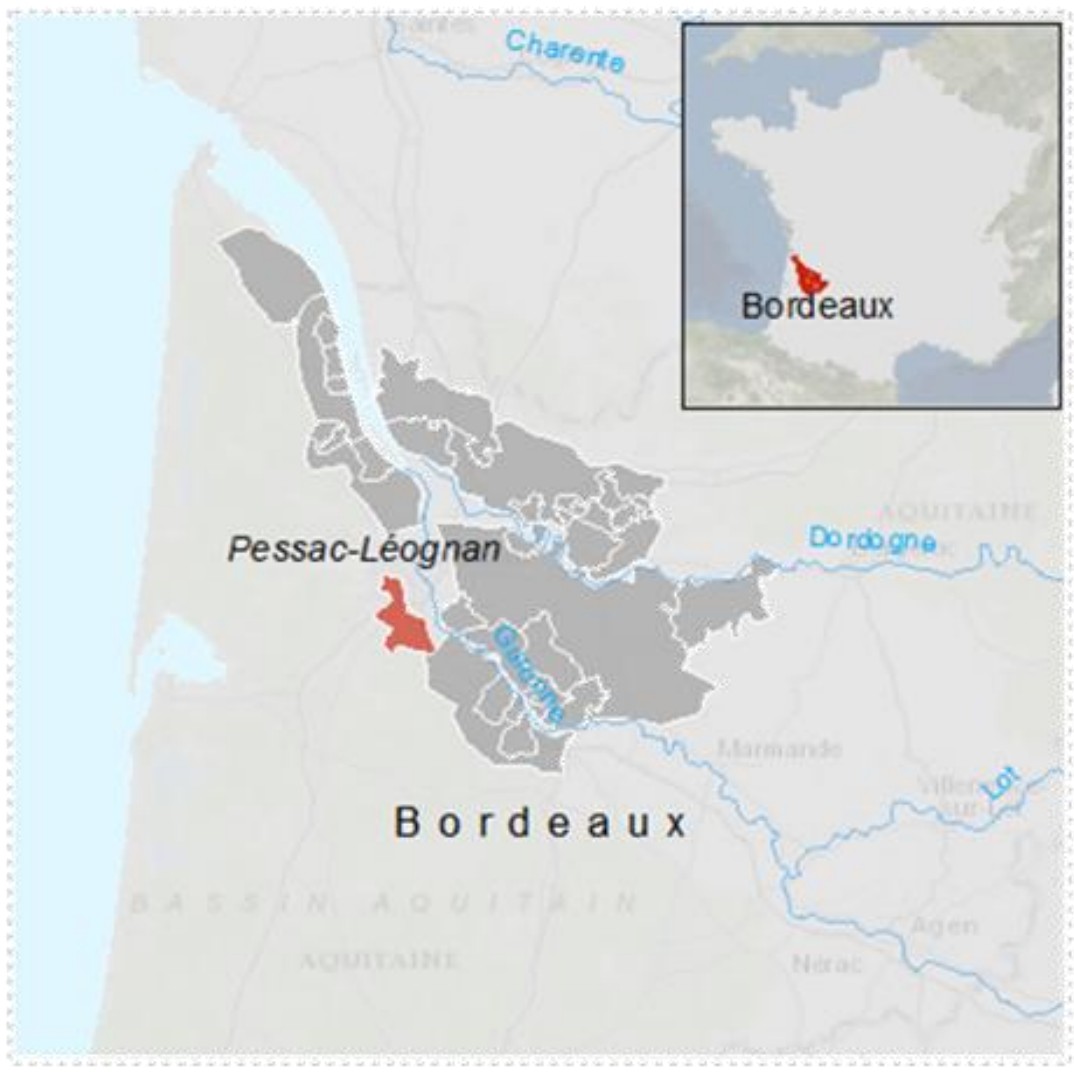

**Figure 1: Location of the experimental site in Bordeaux (France).**

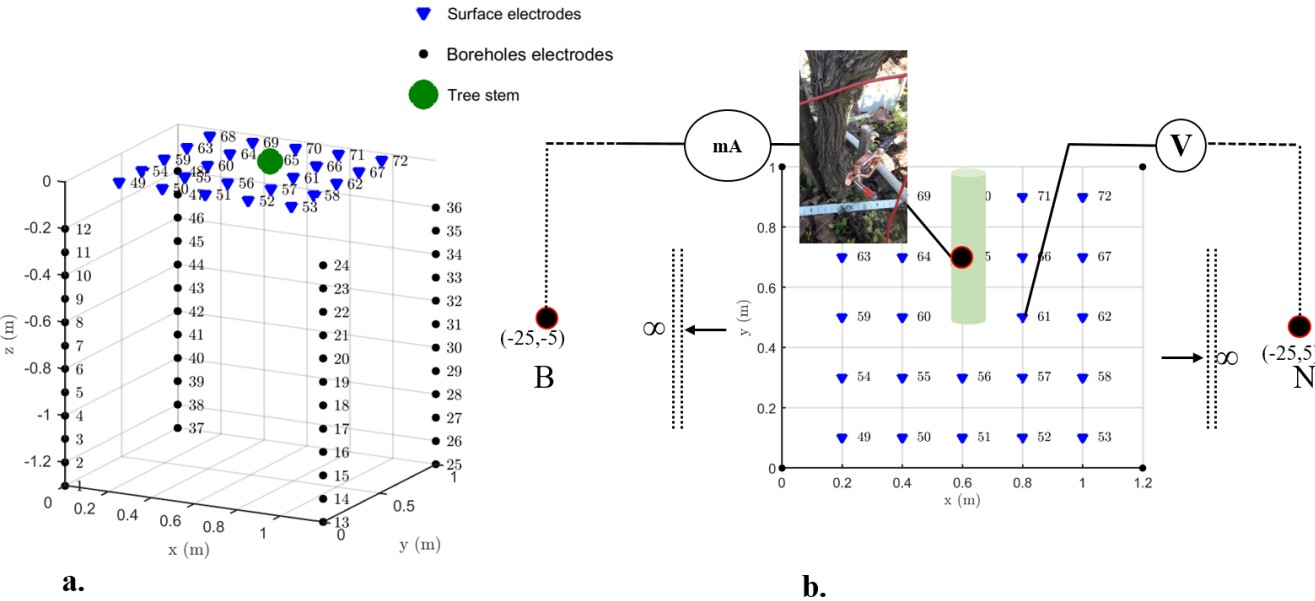

**Figure 2: 3-D schemes of electrical resistivity tomography (ERT) (a) and Mise-à-la-masse (MALM) mesh (b); B and M are remotes electrodes placed 25m apart from the plot.**

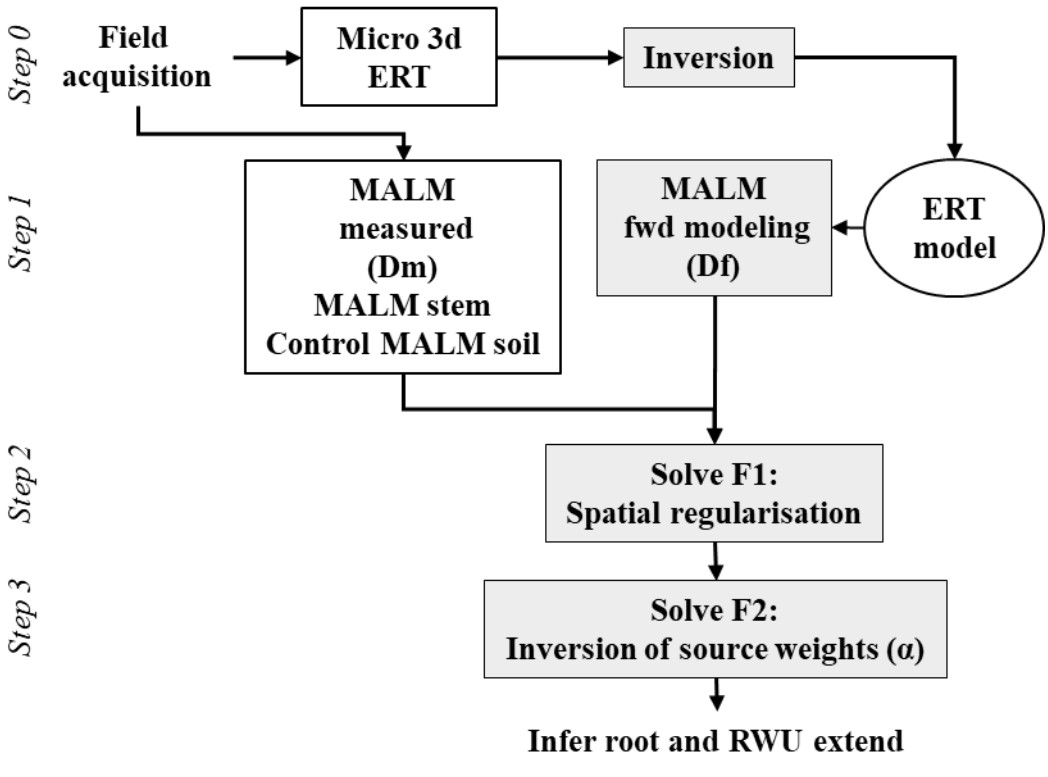

**Figure 3: Flow chart of the analysis of MALM as described in this paper, from data acquisition, processing and interpretation in term of RWU region identification.**

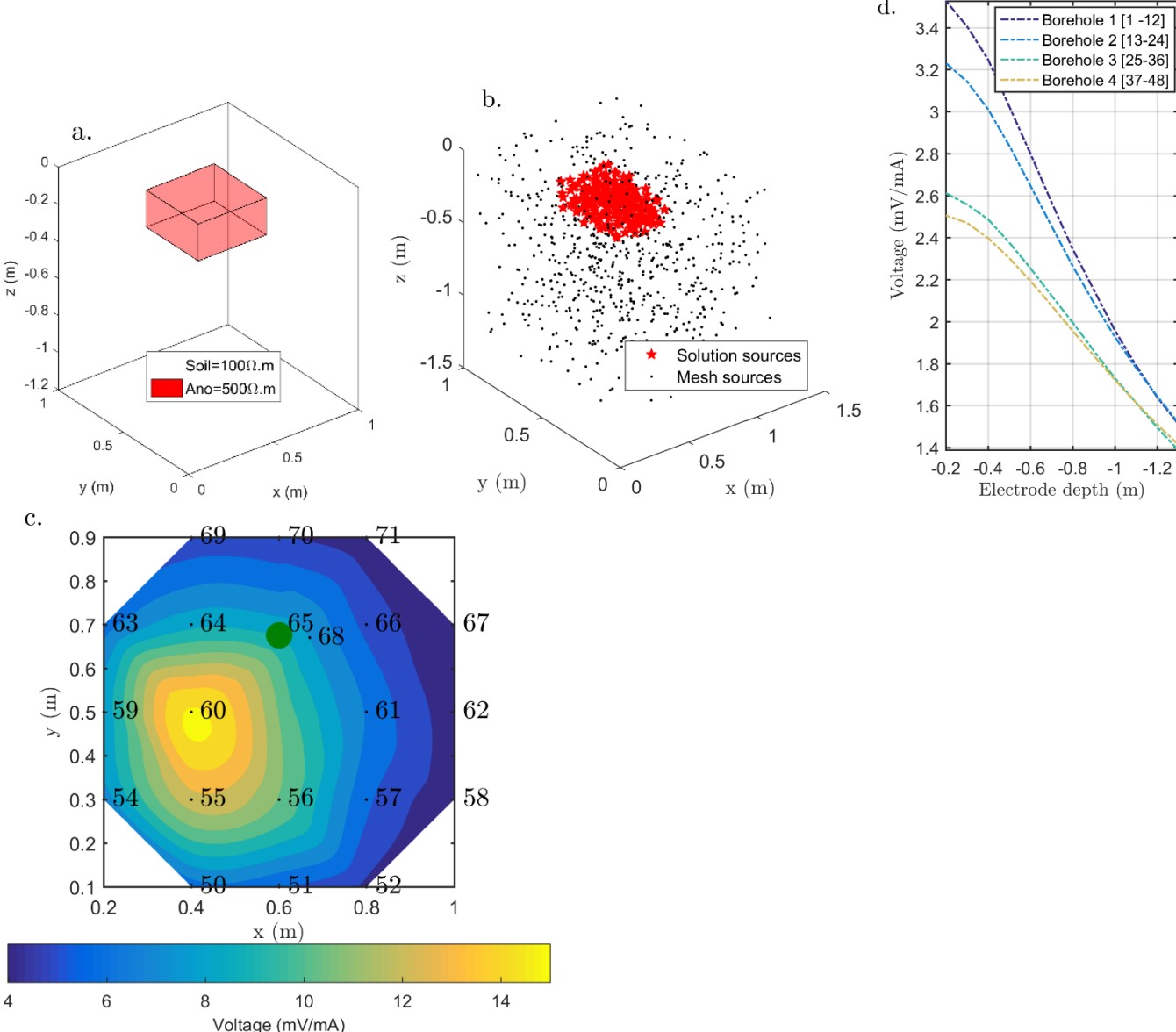

**Figure 4: (a) Initial anomaly of 500 Ω.m located in a domain of lower resistivity (100 Ω.m); (b) Black dots are all virtual sources tested during inversion process, red stars sources forwarded to compute the solution; (c and d) Solution of sum of all sources contribution on surface and with boreholes electrodes; The green point shows the positions of plant stem.**

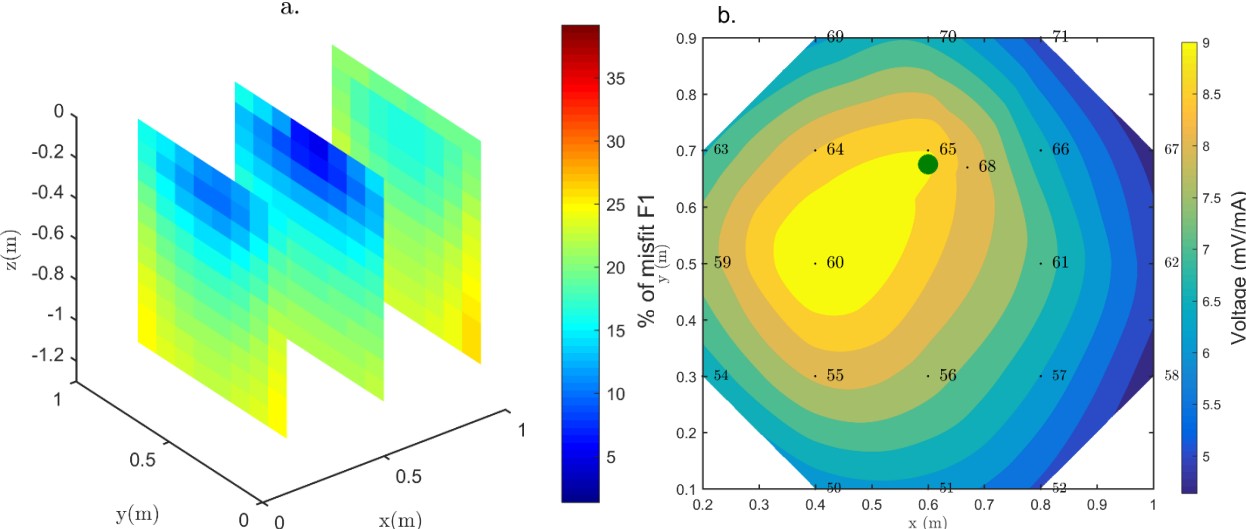

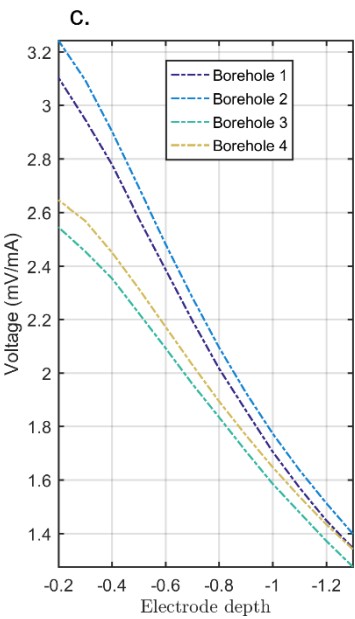

**Figure 5: (a) Spatial distribution of F1: the black dots show the virtual sources locations. In the right top corner, the selected sources (for a misfit of 17%) inferred from the study of the cumulative sum of misfit (or curvature); (b) Inverted model obtained after sources ponderation considering the distributed function F2 for surface electrodes. The green point shows the positions of plant stem (c) for boreholes electrodes.**

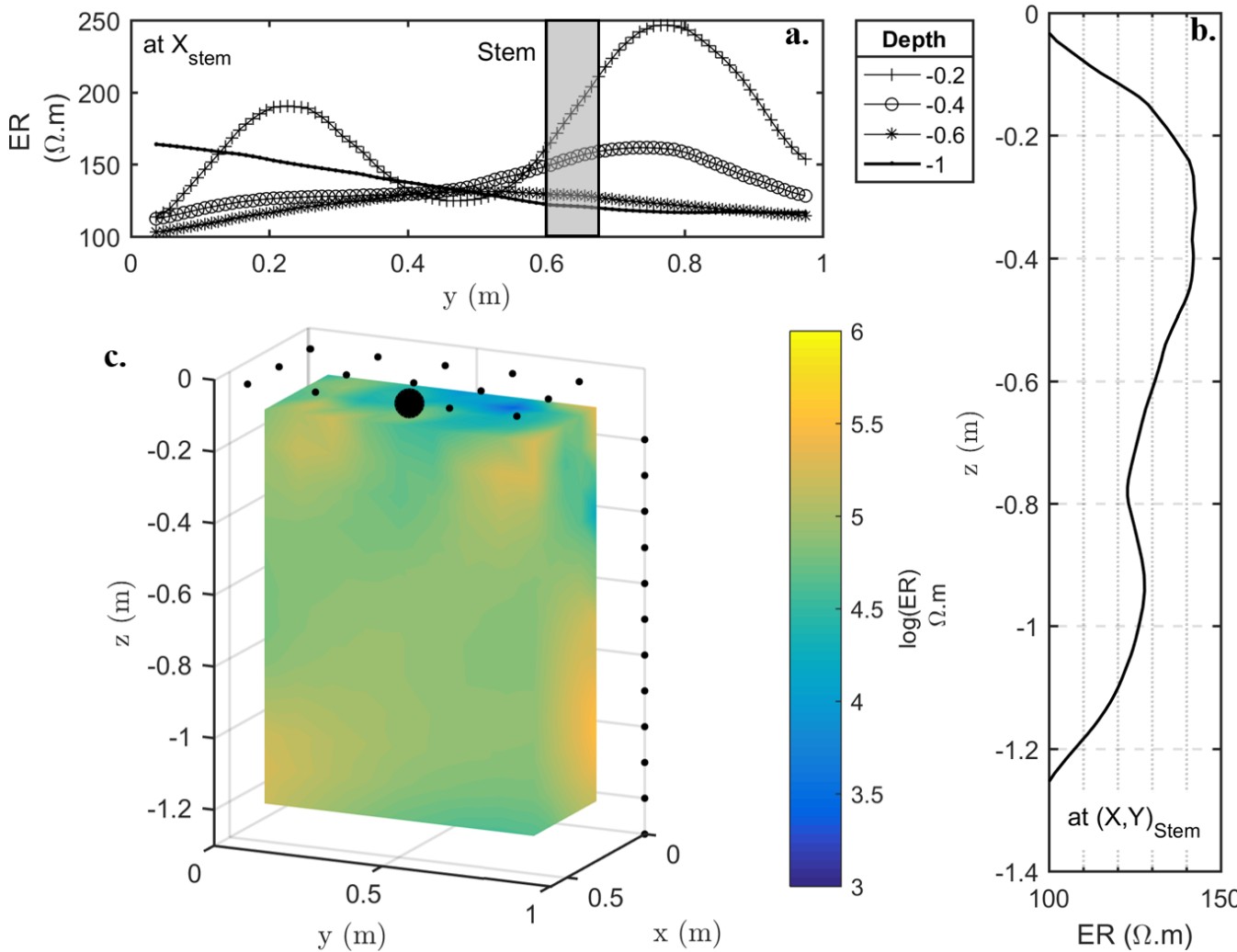

**Figure 6: Results of the 3D ERT inversion: (a) 2D lateral (y direction) variations of resistivity at four depths (0.2 m; 0.4 m; 0.6 m; 1 m); (b) 2D vertical variations of resistivity at the tree stem location; (c) 3D resistivity volume (log scale) sliced at x=0.5 m, with the black point showing the location of the plant stem.**

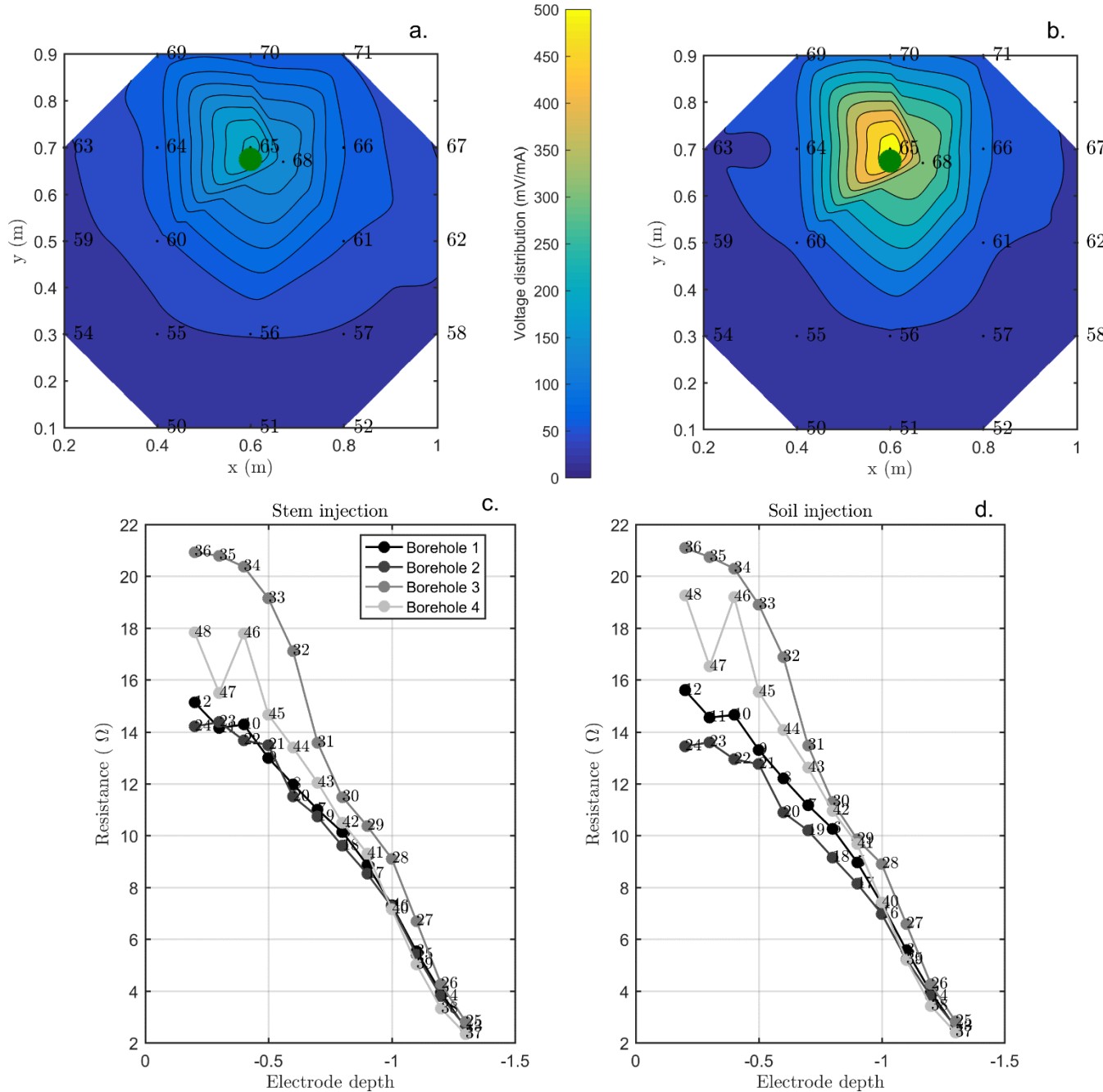

**Figure 7: MALM acquisitions: spatial variations of the normalized voltage (in V/A) observed at surface and borehole electrodes. A comparison is shown between MALM voltage distributions when the current is injected into the soil (b and d) and into the stem (a and c). The green points show the positions of plant stem.**

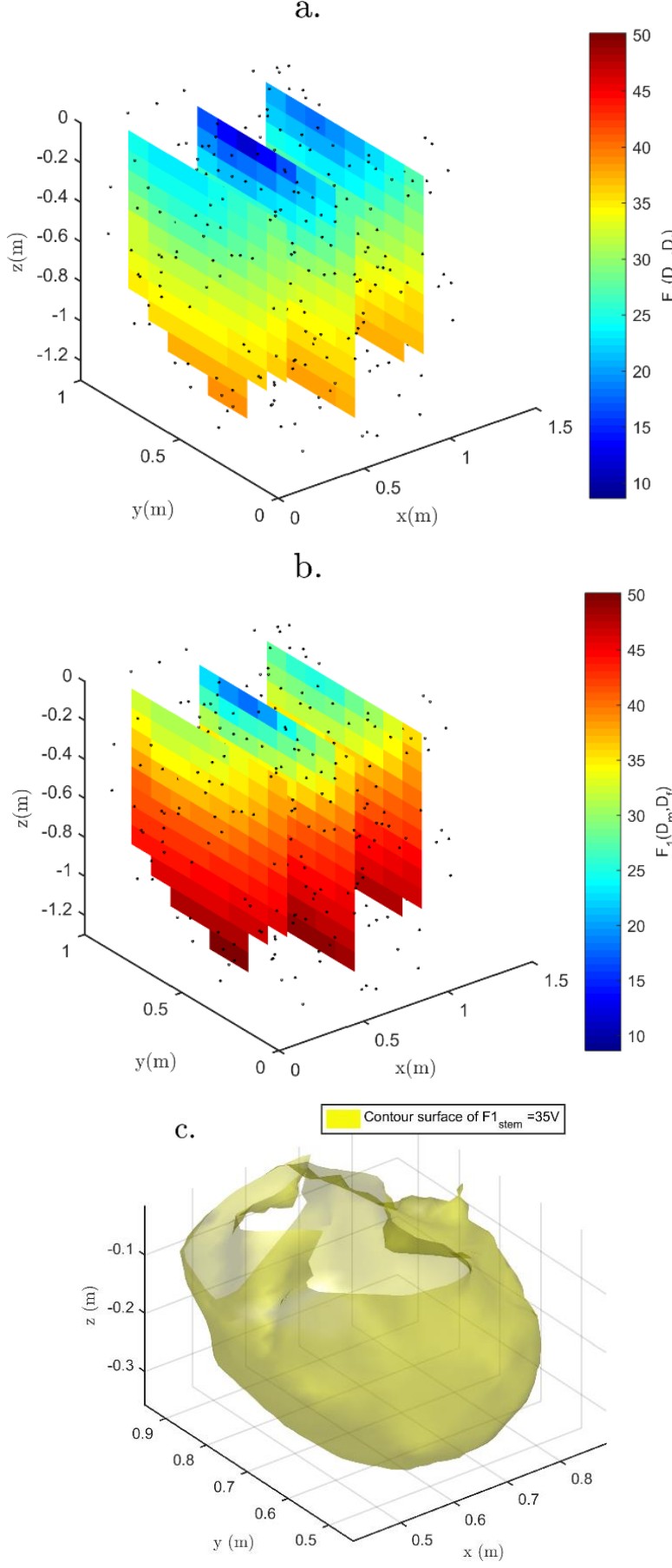

**Figure 8: Spatial distribution of the F1 misfit function e.g. Eq. (1) computed against field data and using the ERT-derived electrical resistivity distribution. (a) shows the case with stem injection, (b) the right panel the case with soil injection, (c) the contour surface of F1=17V in the stem injection case for which only locations that would contribute in a substantial manner to reducing the F1 misfit are used.**

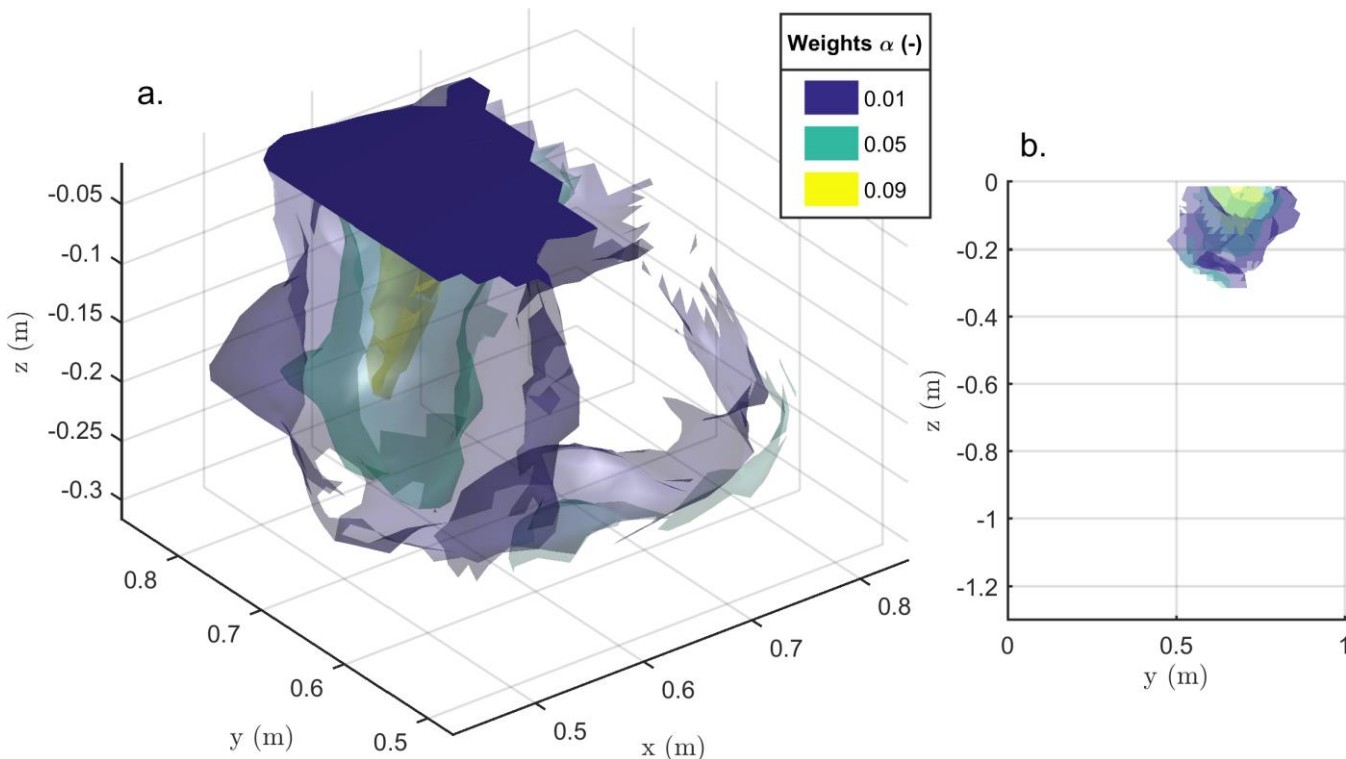

**Figure 9: 3d view (a) and 2d YZ view (b) of the iso-surfaces of current source contribution α after minimization of the objective function F2 as defined in Eq. (2). The results are relevant to stem current injection.**

