# Peer review of "Small scale characterization of vine plant root water uptake via 3D electrical resistivity tomography and Mise-à-la-Masse method."

_Hydrology and Earth System Sciences, 2018_

## Referee Comment (RC1) · Anonymous Referee #1 · 6 Jul 2018

This article describes how electrical resistivity tomography and Mise-à-la-Masse method can be used in conjunction to map root activity or root zone. Indeed, it is a very interesting approach applied to soil-root research for the first time. The authors provides a nice introduction of non-invasive methods of soil root system and then discuss MALM inversion procedure and provide very interesting results indeed. For instance, they are able to map the root extension or root active zone.

In the introduction, it would be nice to also refer the paper Werban et al. (2008) who did interesting ERT study on lupine roots and showed that rooted soil differs from bare soil in terms of pedophysical model.

[Figure]

In P2L27, authors rightly mentions that understanding of contributions from individual root segment on bulk electrical conductivity is limited. I would suggest to refer a recent modeling paper that tries to address exactly this same issue: (Rao et al.: A mechanistic model for electrical conduction in soil–root continuum: a virtual rhizotron study, Biogeosciences Discuss., https://doi.org/10.5194/bg-2018-280, in review, 2018.). In fact, similar modeling study with explicit representation of root structure in the MALM forward modeling should be done as a follow up work to understand better how this approach can be made more robust. Authors could suggest this in the conclusion section.

In P3L29, authors mentions that current injected in stem is most likely to exit the root system only at fine root locations. It would be nice to justify this assumption by providing more information on range of resistance in inner layer of root, outer bark structure as well as fine roots. For instance, current can also exit from woody root radially, if bark layer is anisotropic and offers radially much lower resistance, allowing current to exit even before they reach finer roots. For example, Anderson and Highinbotham (1976) showed that for maize roots, they measured electrical resistance in axial and radial directions of both inner and outer layers of root. They show that radially resistance is much lower for the roots. It would be nice to see, if the above mentioned scenario is discussed in this paragraph and if MALM could be applied to, for instance maize root or non-woody roots or instead can we predict anisotropy in root structure using this approach? However in P11L20, authors also rightly point out that more research is needed to know exactly where current exits the root. In this context, I suggest to discuss the electrical anisotropy of root tissues in the same paragraph and mention how MALM would possibly perform for a highly anisotropic root system. First root is electrically anisotropic at microscopic scale (few centimeter) and also macroscopically the root architecture and soil water uptake pattern induce anisotropy in electrical conductivity. But using MALM to study anisotropy of root structures can itself be an independent study but the authors can suggest that research can take place in this direction using the techniques mentioned in the paper.

Section 2.3.2 wasn't completely clear to me. For example, I had difficulty in understanding the lines 22 to 24 in P7. May be resentencing these lines can bring more information. Although, I get the overall idea of using Eq.[1] and [2] to find secondary current source location in underground, I believe sections 2.3.2 and 2.3.3 still needs some improvement in terms of explanation. For example, the no. of nodes in the mesh and the no. of electrodes are the vector size of Dm and Df, which determines the matrix size of F1 and F2. The values used in the paper can be specified along with computational time it took to do MALM inversion (if it was lengthy process). The term Ns in Eq. [2] is not clearly mentioned and its corresponding value of choice. Is Ns = no. of nodes in the mesh? What authors call sources in these sections are nothing but regions where current (injected by real source in stem) exits the root system (below ground) due to electrical conductivity/ electric potential gradient and hence these are not active sources, so I would suggest to use more physically more meaningful term such as secondary source or passive source to really distinguish it from primary or active source of current that is injected in tree stem. Calling them just source might be misleading (in my opinion).

In general, the explanation for MALM inversion procedure in the paper is not adequate. Efforts could be taken to make it a bit more clear for novice readers. For example, authors mentions MALM inversion as ill-posed, so what kind of regularization, did authors use to make solution stable ?

Secondly, I would suggest some minor suggestions which would, in my opinion, improve the readability of the paper.

• I suggest to use the word non-invasive throughout the journal to have uniformity and the word non-intrusive in P2L8 can be changed to non-invasive. • In P2L18, abbreviation FTSW is not defined. • There is a unwanted comma in (Amato et al. 2009;, ) at P2L22. • In P4L20, stating resistivity is complex number for alternating current may not be so appropriate,in my opinion, as compared to saying resistivity is complex number for polarizing medium having resistance and capacitance parts. Even

for DC current, for example capacitor discharging, we can associate this effect into complex resistivity or impedance. • In L26P6, why is the term micro ERT used, if it is same as 3D ERT which is used throughout, I would remove the term micro to avoid confusion or add it everywhere. • I would add figure indices a), b), c), and d) in Figure A.1 P14 • I would add figure indices a),b),c) along with xlabel,ylabel and SI units of quantities presented in the Figure A.2, P15. • I would add figure indices a) and b) in Figure 2 P23. Also in the left figure, I would put surface electrodes in different color or marker to differentiate it with borehole labels. Once can also change the angle of the tilt in the left figure so that all 4 pillars of boreholes are visible without overlapping. Also size of left figure can be increased by making both the figures as column instead of side-by side row figures. • The underlined term Regularization in Figure 3, P24 seems to be unconnected to flowchart. Rather it should be used inside the blocks where it is used. • Add units to axis labels in Figure 4b P25. The green dot or circle in Fig 4c is not described in caption. Also electrode numbers are not visible. I would also specify electrode numbers corresponding to Borehole1, Borehole2, Borehole3 and Borehole4, so that Figure 4d can be easily compared with Figure 2 a (left) to know the location of these boreholes. • Again green circle in Figure 5 b in P26 needs to be described in caption below. Size of electrode label can be increased for readability. • In Figure 6 b P27, x-axis limits can be reduced from maybe 80 to 150 and adding vertical grids will make the quantitative inspection of variations in soil resistivity, relatively easy. Also size of Figure 6c can be increased. • Figure indices a),b),c), and d) needs to be added in Figure 7, P28. The color contrast between Borehole1 and Borehole 2 is low in bottom figures. • Figure index a) and b) are missing in Figure 9 at P30. • In P6L26, suddenly author uses the new term micro ERT which can be confusing since previously 3D ERT is used. I would suggest to either use micro ERT throughout or use 3D ERT everywhere. • In P7L13, MATLAB version and the name of the optimization function could be mentioned for repeatability reasons. • I would suggest mentioning the units of Dm and Df in Eq 1 and 2. For example, in P7L7, authors can mention Dm is the measured voltage instead of measured data

to be more precise. Also, the word "forwarded" in the same line can be changed to forward voltage data.

I conclude by saying that this paper indeed brings new knowledge in the field of non-invasive imaging of soil-root system and is definitely relevant for publication in HESS. However, I recommend publication after addressing the above listed comments and suggestions.

---

## Referee Comment (RC2) · Anonymous Referee #2 · 21 Aug 2018

Dear Authors,

This article tries to develop a new tool to map and quantify active root distribution using as model plant the grapevine. Thank you for your effort. The problem you are trying to address is very meaningful to a number of disciplines. The combined use of MALM and 3D ERT is very novel for this purpose, it is state of art and it was worth trying.

You are addressing a very multidisciplinary problem, but the plant science/agriculture/soil science section is lacking, from a practical (no data on roots) and conceptual point of view (the assumption that only fine roots could conduce). Claiming that a tree such as the grapevine would have such a shallow rooting depth, it is

very, very hard to believe, and is in contrast with the published literature and not in agreement as you state.

The paper is very advanced from the geophysical perspective, however it sounds more like the presentation of preliminary data then the report of an accomplished work. The text continuously refers to roots, and the goal of the paper is mapping roots, but there is not a single figure showing a root map obtained through the method, only voltages and misfits. Authors do not bring evidence that their image is root related, why it could not be just noise? Why the root system was not excavated to confirm your imaging? Why there are no measurements on the root system at all?

The article is well written, figures are of good quality, but in my opinion the article cannot be published in the current form. Bringing root data would oblige repeating the whole experiment, as roots of the measured plant would have changed in between. In this case, author should take into account that measuring a single plant would not be enough, because of biological variability. Without a new field campaign, the text should be completely rewritten. Authors should avoid to speak of roots if not as the ultimate research goal in order to be coherent with the data presented which are only of geophysical nature. However, once rewritten from an exclusively geophysical point of view the text will be very technical and much less interesting to the wide audience of this journal.

Abstract:

The abstract is very generic. It does not give clear indication of material and methods, neither of results. The evidence from field data: it is not clear why the voltage distribution support the hypothesis, as current is injected in two different media. "in agreement with literature on similar crops" is simply not enough, authors should have measured the plant under investigation, and also repeated the experiment with more than one plant to deal with the biological variability.

Introduction:

The stated problem is very wide and generic. This paper deals with the development of a new method for the 3D mapping of fine roots, which constitutes the most active part of a root system from a hydrological and biological point of view.

I would not call MALM non-invasive, as a stainless steel rod needs to be inserted in the stem, and what is presented in picture 2 does not look as a small intervention either. Would a steel ring be adaptable to the technique?

Section 1.2. This section reviews attentively the precedent literature on the use of geophysical methods for morphological monitoring of roots, however some more is needed on the use of geophysical methods for observing root activity (nutrient and/or water uptake) which is notably neglected in this section.

P2L13. clay content: the Archie law does not really account for clay content, conversely the matrix is supposed non-conductive, Waxman and Smits were probably the first authors to account for this. Temperature also is not present in the Archie law, and the cementation exponent is supposed independent from temperature effect. Generally the study of the effect of temperature is reported to

Campbell, R.B., Bower, C.A., Richards, L.A., 1948. Change of electrical conductivity with temperature and the relation of osmotic pressure to electrical conductivity and ion concentration for soil extracts. Soil Sci. Soc. Am. Proc. 13, 66–69.

P2L14. In my opinion the use of the term pedotransfer functions here is wrong. Those are petro-physical (mechanistic) models. You could use pedo-physical, but these works were done for geological more than soil investigation. Recent works on electrical pedo-transfer functions also including the relationships with soil properties you are interested in, may be found in:

Hadzick, Z.Z., Guber, A.K., Pachepsky, Y.A., Hill, R.L., 2011. Pedotransfer functions in soil electrical resistivity estimation. Geoderma 164, 195–202.

Brillante, L., Bois, B., Mathieu, O., Bichet, V., Michot, D., Lévêque, J., 2014. Monitoring soil volume wetness in heterogeneous soils by electrical resistivity. A field-based pedotransfer function. J. Hydrol. 516, 55–66.

P2L17. acronyms must be capitalized when written in full at first appearance

P2L19. There are works from the same authors for the mapping of tree roots, that actually came first. However, their method in my personal experience has been difficult to replicate. Especially in heterogeneous soil-moisture conditions that are the frequently the conditions of these studies. Furthermore, the method is very labor intensive and destructive, as it demands a direct calibration with roots from soil cores.

Amato, M., Basso, B., Celano, G., Bitella, G., Morelli, G., Rossi, R., 2008. In situ detection of tree root distribution and biomass by multi-electrode resistivity imaging. Tree Physiol. 28, 1441–1448.

Rossi, R., Amato, M., Bitella, G., Bochicchio, R., Ferreira Gomes, J.J., Lovelli, S., Martorella, E., Favale, P., 2011. Electrical resistivity tomography as a non-destructive method for mapping root biomass in an orchard. Eur. J. Soil Sci. 62, 206–215. https://doi.org/10.1111/j.1365-2389.2010.01329.x

P2L19-20. The cited work from Cassiani does not report resistivity∼root correlation/calibration. How could this paper demonstrate the reliability of the method as in Amato and coworkers?

P2L23-26. is very interesting and you could also expand considering the case of soil with heterogeneous moisture or soil properties that would complicate further the calibration with root mass

P2L32. It is not clear if you are speaking of the resistivity of roots in soil (and at this point there is also the effect of the mass), or of a root fragment in itself.

The term dead wood is not correct, you could just use "heartwood and the pith"

P3L4-6. citations needed.

P3-L25. "sap flow processes take place at the cambium" is not correct, sap flow happens in xylem and phloem which are from either side of the cambium.

P3L27-28. "the quasi-infinite connection" is this your assumption? In this case needs to be clearly stated, or you could cite the paper that addresses this.

P3L29-30. This may eventually be considered a strength of the method, as fine roots are the active ones, people in plant sciences would be interested to measure. You could put this more in evidence.

P4L8. xylem and phloem sap, as xylem and phloem are not fluids. P4L27 Maybe this citation would be interesting for future readers:

Postic, F., Doussan, C., 2016. Benchmarking electrical methods for rapid estimation of root biomass. Plant Methods 12:33. https://doi.org/10.1186/s13007-016-0133-7

Material and methods:

The biological material and agricultural system needs to be better described (how old are the plants, what is the distance between them, what shape size the canopy, what is the rootstock?)

2.1.1 Interpreted soil picture needed

P3L11. "colluviosol" is not international nomenclature. Please correct using IUSS Working Group WRB, 2014. World reference base for soil resources 2014. International soil classification system for naming soils and creating legends for soil maps, World Soil. ed. FAO, Rome

P3L11. Measurement of root density is key in this paper. How this was performed needs to be detailed.

P3L13. What is "dense rooting for you?" please specify

P3L14. The term horizon instead of layer would be more scientific (from now on)

P3L17. "terroir de grave" not clear what this is. Is this a term for a distinctive geological or pedological formation that is meaningful in local terminology? If this is a local term needs to be reported in italics and quoted

P3L18. Only one plant??? That is far under the minimum for characterizing a biological systems, and even if you just want to show that the method is working how could you claim that is not just by chance?

P3L19. "sandy-clayey horizon" although correctly identifying and naming it would be better

P3L21. English not clear (top layer rooting). Do you mean "asymmetric root development in the top layer"?

Sections 2.2 and 2.3 In agreement with Reviewer 1 the geophysical acquisition and inversion procedure is not detailed enough to be replicated by new authors. This must be better detailed. Are you considering to share the MALM inversion code with this article?

Results:

P10L9-11. A root system with a lateral extension of 0.5-0.9m appears reasonable (considering the density of plants in figure 2), however no active roots deeper than 0.3m, in a non-irrigated permanent tree it is a very striking and conflicting result. In my opinion this is not true, as I am not even sure that in a tilled soil, with grass in the inter row (figure 2) you will find the majority of tree roots in the first 0.3m. The problem here is that authors did not excavate the rootzone to demonstrate their measurements. Very few examples are reported here:

Water depletion and contribution to plant water status is observed under grapevines at 1.5m using electrical resistivity tomography in Brillante, L., Bois, B., Lévêque, J., & Mathieu, O. (2016). Variations in soil-water use by grapevine according to plant water status and soil physical-chemical characteristics-A 3D spatio-temporal analysis.

European Journal of Agronomy, 77, 122–135.

Old-school maps of grapevine roots down to a depth of 1.5m (in irrigated conditions) can be find in Rob M. Stevens, Tim Douglas Distribution of grapevine roots and salt under drip and full-ground cover microjet irrigation systems. Irrig Sci (1994) 15:147-152

Using magnetic resonance imaging root activity is already observed at around 30cm depth in just 20 days old Lupinus alba seedlings, (Carminati A (2013) Rhizosphere wettability decreases with root age: a problem or a strategy to increase water uptake of young roots? Front. Plant Sci. 4:298. doi: 10.3389/fpls.2013.00298)

Using modelling and isotopic approaches the mean relative contribution to transpiration of soil layers under 0.4m is estimated around 50% in Rothfuss and Javaux, 2017 Reviews and syntheses: Isotopic approaches to quantify root water uptake: a review and comparison of method Biogeosciences, 14, 2199–2224, 2017

P10L15-16. This work does not show that MALM can provide anything useful to discriminate spatial distribution of roots if not lateral extension within the first 30cm of soil depth.

P10L27-30. There is no evidence in this article "that injecting current in the plant stem causes a distribution of electrical current sources in the ground that correspond to the location of active roots." Although I would eventually agree with this author suggestion, there is not theoretical argumentation to sustain this in this text, neither experimental evidence.

P11L5-14. This is well written

P11L17-19. This is not true, as also through the bark there can be some, small water uptake, which could allow electrical flow. Cuneo et al., 2018, Water uptake can occur through woody portions of roots and facilitates localized embolism repair in grapevine. New Phytologist, 218(2)

P11L19-22. Interesting field of research

P11L24-25. Slightly narrower range?? See previous comment, also when referring to literature, please cite at least one article!

P11L25-26. Out of context. No data to support the sentence. Remove.

P11L33. This is sure

P12L8. Interesting field of research

Conclusion:

Avoid to repeat the methods in the conclusion

---

## Editor Comment (EC1) · M Vanclooster (Editor) · 21 Aug 2018

Dear Authors,

Your manuscript HESS-2018-238 received 2 detailed referee reports. Both referees appreciated the novelty of the research but regretted that some methodological details are lacking.

The second referee identified also a set of concerns related to the plant physiological and biological aspects of the experiment. He/she also strongly suggested having an independent observation of root mass distribution to validate the indirect assessment

as made with the MALM. I'm afraid that this last concern cannot be considered since this would mean a completely new experimental set-up.

Given the novelty of the research, I suggest incorporating the many detailed remarks in a major revision of the paper. I also suggest introducing a more detailed description of the methodological aspects and a discussion on how the validation can be strengthened in the future by independent root mass distribution observations.

I look forward to receiving a major revised version of this interesting manuscript.

Sincerely yours,

———————————————————

---

## Author Comment (AC1) · 21 Sep 2018

We thank the Reviewer for his/her comments. In the ensuing text, we try and address all raised issues. The reviewer's comments are reported in black, our replies in *italic blue.*

This article describes how electrical resistivity tomography and Mise-à-la-Masse method can be used in conjunction to map root activity or root zone. Indeed, it is a very interesting approach applied to soil-root research for the first time. The authors provides a nice introduction of non-invasive methods of soil root system and then discuss MALM inversion procedure and provide very interesting results indeed. For instance, they are able to map the root extension or root active zone.

- In the introduction, it would be nice to also refer the paper Werban et al. (2008) who did interesting ERT study on lupine roots and showed that rooted soil differs from bare soil in terms of pedophysical model.

*One sentence was added to describe Werban et al. (2008)'s work: "Werban et al. (2008) performed an interesting ERT study on lupine roots and showed that rooted soil differs from bare soil in terms of pedo-physical model", see page 2.*

- In P2L27, authors rightly mentions that understanding of contributions from individual root segment on bulk electrical conductivity is limited. I would suggest to refer a recent modeling paper that tries to address exactly this same issue: (Rao et al.: A mechanistic model for electrical conduction in soil–root continuum: a virtual rhizotron study, Biogeosciences Discuss., https://doi.org/10.5194/bg-2018-280, in review, 2018.).

*Thanks for the suggested literature reference. We introduced Rao et al. work in our literature review.*

- In fact, similar modeling study with explicit representation of root structure in the MALM forward modeling should be done as a follow up work to understand better how this approach can be made more robust. Authors could suggest this in the conclusion section.

*Thanks for the suggestion. We are also currently working on a modelling approach and it is for sure a good perspective. One sentence was added in the conclusion section: "Modelling study with an explicit representation of root structure in the MALM forward modelling may be done as a follow up work to understand how the proposed approach can be made more robust."*

- In P3L29, authors mentions that current injected in stem is most likely to exit the root system only at fine root locations. It would be nice to justify this assumption by providing more information on range of resistance in inner

layer of root, outer bark structure as well as fine roots. For instance, current can also exit from woody root radially, if bark layer is anisotropic and offers radially much lower resistance, allowing current to exit even before they reach finer roots. For example, Anderson and Highinbotham (1976) showed that for maize roots, they measured electrical resistance in axial and radial directions of both inner and outer layers of root. They show that radially resistance is much lower for the roots. It would be nice to see, if the above mentioned scenario is discussed in this paragraph and if MALM could be applied to, for instance maize root or non-woody roots or instead can we predict anisotropy in root structure using this approach?

However in P11L20, authors also rightly point out that more research is needed to know exactly where current exits the root. In this context, I suggest to discuss the electrical anisotropy of root tissues in the same paragraph and mention how MALM would possibly perform for a highly anisotropic root system. First root is electrically anisotropic at microscopic scale (few centimeter) and also macroscopically the root architecture and soil water uptake pattern induce anisotropy in electrical conductivity. But using MALM to study anisotropy of root structures can itself be an independent study but the authors can suggest that research can take place in this direction using the techniques mentioned in the paper.

*We added one sentence in the introduction concerning Anderson and Highinbotham results, and a longer paragraph in the Discussion section to discuss root anisotropy and a possible use of MALM to study it.*

- Section 2.3.2 wasn't completely clear to me. For example, I had difficulty in under- standing the lines 22 to 24 in P7. May be resentencing these lines can bring more information.

*The paragraph has been partly resentenced for clarification. A references to Figure 3 was introduced to clarify the sentence:*

*"[…] the resistivity distribution obtained from ERT was used, as necessary in real cases, as the background resistivity through which the current, induced by the MALM experiment, flows. In this synthetic example, the MALM datasets (see Fig. 3 – ERT Model) are obtained hypothesizing current source locations (at the FE mesh nodes) within the given theoretical "root" zone – the current intensity is assumed the same at all nodes."*

- Although, I get the overall idea of using Eq.[1] and [2] to find secondary current source location in underground, I believe sections 2.3.2 and 2.3.3 still needs some improvement in terms of explanation. For example, the no. of nodes in the mesh and the no. of electrodes are the vector size of Dm and Df, which determines the matrix size of F1 and F2. The values used in the paper can be specified along with computational time it took to do MALM inversion (if it was lengthy process).

*We introduced more details to describe and discuss the number of nodes in the mesh:*

*[…] "The number of current sources to invert for (Ns) is primarily dictated by the desired input mesh quality (Fig.A2c). This is determined by the required computational time. For this small scale prospection, we adopted a mesh composed of 23700 nodes (including remotes electrodes, e.g. Fig.A2a). The inversion region was limited to 3618 nodes (Fig.A2b). Furthermore, as shown in Fig. 3, the strategy is to use the F1 and F2 optimizations sequentially. In order to guide the physically sound F2 inversion, initial values of $\alpha_{io} = [\alpha_{1o}, \alpha_{2o}, …, \alpha_{Nso}]$ were set using normalized F1 values (between 0 and 1). This is equivalent to applying a regularization based upon the initial F1*

*search upon the F2 optimization. A global optimization using a Constrained Nonlinear Optimization Algorithms (fmincon solver using a gradient-based method associated with the Sequential quadratic programming (SQP) optimization algorithm) method implemented in MATLAB® (R2016b) software was then used to minimize F2."*

- The term Ns in Eq. [2] is not clearly mentioned and its corresponding value of choice. Is Ns = no. of nodes in the mesh?

*Ns is now specified in the text.*

- What authors call sources in these sections are nothing but regions where current (injected by real source in stem) exits the root system (below ground) due to electrical conductivity/ electric potential gradient and hence these are not active sources, so I would suggest to use more physically more meaningful term such as secondary source or passive source to really distinguish it from primary or active source of current that is injected in tree stem. Calling them just source might be misleading (in my opinion).

*We partly disagree with the reviewer. Here calling the current exiting the roots "Secondary sources" can be confusing in the scope of electrical measurements since this terminology relates commonly to the induced polarization method (i.e. a source proportional to both the intrinsic chargeability and the primary (applied) current density).*

- In general, the explanation for MALM inversion procedure in the paper is not adequate. Efforts could be taken to make it a bit more clear for novice readers. For example, authors mentions MALM inversion as ill-posed, so what kind of regularization, did authors use to make solution stable?

*We have tried to clarify how the inversion procedure has been performed. Indeed, there is no formal "regularization" in the sense e.g. of an Occam's approach. Yet, there are two "regularizing" steps during the whole process: the first is a "spatial regularization" (as called in the paper): this is obtained via the F1 function, which selects an ensemble of candidate source locations. The second is implicit in the gradient based solver (fmincon) associated with the SQP algorithm use in the F2 inversion.*

*This two regularization steps are now fully explained in section 2.3.2*

Secondly, I would suggest some minor suggestions which would, in my opinion, improve the readability of the paper.

- I suggest to use the word non-invasive throughout the journal to have uniformity and the word non-intrusive in P2L8 can be changed to non-invasive

*Corrected.*

- In P2L18, abbreviation FTSW is not defined.

*FTSW= Fraction of Transpirable Soil Water; corrected in the text.*

- There is a unwanted comma in (Amato et al. 2009;, ) at P2L22.

*Corrected.*

- In P4L20, stating resistivity is complex number for alternating current may not be so appropriate, in my opinion, as compared to saying resistivity is complex number for polarizing medium having resistance and capacitance parts. Even for DC current, for example capacitor discharging, we can associate this effect into complex resistivity or impedance.

*Corrected.*

- In L26P6, why is the term micro ERT used, if it is same as 3D ERT which is used throughout, I would remove the term micro to avoid confusion or add it everywhere.

*Corrected.*

- I would add figure indices a), b), c), and d) in Figure A.1 P14

*Done*

- I would add figure indices a),b),c) along with xlabel,ylabel and SI units of quantities presented in the Figure A.2, P15.

*Done*

- I would add figure indices a) and b) in Figure 2 P23. Also in the left figure, I would put surface electrodes in different color or marker to differentiate it with borehole labels. Once can also change the angle of the tilt in the left figure so that all 4 pillars of boreholes are visible without overlapping. Also size of left figure can be increased by making both the figures as column instead of side-by side row figures.

*Done*

- The underlined term Regularization in Figure 3, P24 seems to be unconnected to flowchart. Rather it should be used inside the blocks where it is used.

*Done*

- Add units to axis labels in Figure 4b P25. The green dot or circle in Fig 4c is not described in caption. Also electrode numbers are not visible. I would also specify electrode numbers corresponding to Borehole1, Borehole2, Borehole3 and Borehole4, so that Figure 4d can be easily compared with Figure 2 a (left) to know the location of these boreholes.

*Done*

- Again green circle in Figure 5 b in P26 needs to be described in caption below. Size of electrode label can be increased for readability

*Done*

- In Figure 6 b P27, x-axis limits can be reduced from maybe 80 to 150 and adding vertical grids will make the quantitative inspection of variations in soil resistivity, relatively easy. Also size of Figure 6c can be increased.

*Done*

- Figure indices a),b),c), and d) needs to be added in Figure 7, P28. The color contrast between Borehole1 and Borehole 2 is low in bottom figures.

*Done*

- Figure index a) and b) are missing in Figure 9 at P30.

*Done*

- In P6L26, suddenly author uses the new term micro ERT which can be confusing since previously 3D ERT is used. I would suggest to either use micro ERT throughout or use 3D ERT everywhere.

*Done*

- In P7L13, MATLAB version and the name of the optimization function could be mentioned for repeatability reasons.

*We have added such details: A global optimization was conducted (for F2 minimization) using a Constrained Nonlinear Optimization Algorithms gradient based algorithm (the fmincon function associated with the SQP algoritm) as implemented in MATLAB® (R2016b).*

- I would suggest mentioning the units of Dm and Df in Eq 1 and 2. For example, in P7L7, authors can mention Dm is the measured voltage instead of measured data C4 to be more precise. Also, the word "forwarded" in the same line can be changed to forward voltage data.

*Changes made accordingly*

I conclude by saying that this paper indeed brings new knowledge in the field of non- invasive imaging of soil-root system and is definitely relevant for publication in HESS. However, I recommend publication after addressing the above listed comments and suggestions.

*Please also note the revised manuscript red marked as a supplement to this comment*

[revised manuscript text omitted]

---

## Author Comment (AC2) · 21 Sep 2018

We thank the Reviewer for his/her comments. In the ensuing text, we try and address all raised issues. The reviewer's comments are reported in black, our replies in *italic blue*.

Dear Authors,

This article tries to develop a new tool to map and quantify active root distribution using as model plant the grapevine. Thank you for your effort. The problem you are trying to address is very meaningful to a number of disciplines. The combined use of MALM and 3D ERT is very novel for this purpose, it is state of art and it was worth trying.

You are addressing a very multidisciplinary problem, but the plant science/agriculture/soil science section is lacking, from a practical (no data on roots) and conceptual point of view (the assumption that only fine roots could conduce). Claim- ing that a tree such as the grapevine would have such a shallow rooting depth, it is very, very hard to believe, and is in contrast with the published literature and not in agreement as you state.

*We must stress that the vine is a very small tree, particularly in this case. The tree height is less than one 1 meter.*

The paper is very advanced from the geophysical perspective, however it sounds more like the presentation of preliminary data then the report of an accomplished work. The text continuously refers to roots, and the goal of the paper is mapping roots, but there is not a single figure showing a root map obtained through the method, only voltages and misfits. Authors do not bring evidence that their image is root related, why it could not be just noise? Why the root system was not excavated to confirm your imaging? Why there are no measurements on the root system at all?

*We must stress that the conviction that excavating is a good way of showing the roots and their functioning is a wrong one. Hair roots are destroyed in the excavation. Showing the woody roots is showing for the most part the structural support of the tree. We cannot overstress this. Otherwise, excavating using an air spade and: problem solved. This is not true. RWU is controlled by fine structures that are of course in connection with the woody roots, but do not coincide with them. Should this not be the case, this paper would make no sense.*

*Nevertheless, we take the reviewer's opinion as a solid one in the sense that some link to direct investigation is preferred. In the present case, excavation is not possible, as the investigation is*

*made in a privately owned vineyard, and some limitations in direct investigation are to be expected.*

The article is well written, figures are of good quality, but in my opinion the article cannot be published in the current form. Bringing root data would oblige repeating the whole experiment, as roots of the measured plant would have changed in between.  In this case, author should take into account that measuring a single plant would not be enough, because of biological variability. Without a new field campaign, the text should be completely rewritten. Authors should avoid to speak of roots if not as the ultimate research goal in order to be coherent with the data presented which are only of geophysical nature. However, once rewritten from an exclusively geophysical point of view the text will be very technical and much less interesting to the wide audience of this journal.

*We partly disagree with the reviewer's viewpoint, in his/her belief that investigating the roots' functioning could be assessed by excavating the plant off. Also, it is unclear what the reviewer means by the effect of biological variability, as indeed we are trying to push research toward the point of understanding how biological systems react to physical constraints. And yet, with no methods for effective investigations how would we proceed. Are direct methods the only way possible. Would the reviewer undergo destructive surgery with no x-ray CT-scan before?*

**Abstract**:

The abstract is very generic. It does not give clear indication of material and methods, neither of results. The evidence from field data: it is not clear why the voltage distribu- tion support the hypothesis, as current is injected in two different media. "in agreement with literature on similar crops" is simply not enough, authors should have measured the plant under investigation, and also repeated the experiment with more than one plant to deal with the biological variability.

*We partly disagree, as the reviewer is more generic than us in his/her complaint. In which direction shall be move? Yet we tried to improve the Abstract readability and information content.*

**Introduction**:

The stated problem is very wide and generic.  This paper deals with the development  of a new method for the 3D mapping of fine roots, which constitutes the most active part of a root system from a hydrological and biological point of view.

I would not call MALM non-invasive, as a stainless steel rod needs to be inserted in the stem, and what is presented in picture 2 does not look as a small intervention either. Would a steel ring be adaptable to the technique?

*Again the reviewer's comments are more generic than our text. What is the specific suggestion here?*
*Also, the argument about a geophysical method being or not being non-invasive is pointless. Are x-rays non- invasive because one does not stick a nail in the system? Yet x rays can kill…*
*What we mean here is non-destructive*

Section 1.2. This section reviews attentively the precedent literature on the use of geophysical methods for morphological monitoring of roots, however some more is needed on the use of geophysical methods for observing root activity (nutrient and/or water uptake) which is notably neglected in this section.

*We added 4 new references concerning RWU and geophysics within a new paragraph:*
*"Amato et al. (2009) tested the capability of 3-D ERT to quantify root biomass on herbaceous plants using resistivity root correlation/calibration. Electrical methods have been also used to identify Root Water Uptake (RWU - e.g. Cassiani et al., 2012; Garré et al., 2011; Michot et al., 2003; Srayeddin and Doussan, 2009) and demonstrated the match between soil water content variations and temporal changes in electrical resistivity. Cassiani et al. 2016 monitored the electrical resistivity in an apple orchard under external forcing conditions (irrigation and plant driven evaporation) and showed that the increase of resistivity is located in the subsoil region where active roots are present."*

P2L13. clay content: the Archie law does not really account for clay content, conversely the matrix is supposed non-conductive, Waxman and Smits were probably the first authors to account for this. Temperature also is not present in the Archie law, and the cementation exponent is supposed independent from temperature effect. Generally the study of the effect of temperature is reported to

Campbell, R.B., Bower, C.A., Richards, L.A., 1948. Change of electrical conductivity with temperature and the relation of osmotic pressure to electrical conductivity and ion concentration for soil extracts. Soil Sci. Soc. Am. Proc. 13, 66–69.

*We do agree that the reference to Archie's Law does not account for changes of temperature neither for clay content. The sentence has been rephrased putting the references at the right place and a reference to Campbell et al. has been added.*

P2L14. In my opinion the use of the term pedotransfer functions here is wrong. Those are petro-physical (mechanistic) models. You could use pedo-physical, but these works were done for geological more than soil investigation. Recent works on electrical pedo-transfer functions also including the relationships with soil properties you are interested in, may be found in:

Hadzick, Z.Z., Guber, A.K., Pachepsky, Y.A., Hill, R.L., 2011. Pedotransfer functions in soil electrical resistivity estimation. Geoderma 164, 195–202.

Brillante, L., Bois, B., Mathieu, O., Bichet, V., Michot, D., Lévêque, J., 2014. Monitoring soil volume wetness in heterogeneous soils by electrical resistivity. A field-based pedotransfer function. J. Hydrol. 516, 55–66.

*We do agree. The word "Pedotransfer" has been deleted to try and avoid confusion and replaced "by pedo-physical".*

P2L17. acronyms must be capitalized when written in full at first appearance

*Done. E.g.: Fraction of Transpirable Soil Water (FTSW)*

P2L19. There are works from the same authors for the mapping of tree roots, that ac- tually came first. However, their method in my personal experience has been difficult to replicate. Especially in heterogeneous soil-moisture conditions that are the frequently the conditions of these studies. Furthermore, the method is very labor intensive and destructive, as it demands a direct calibration with roots from soil cores.

Amato, M., Basso, B., Celano, G., Bitella, G., Morelli, G., Rossi, R., 2008. In situ detection of tree root distribution and biomass by multi-electrode resistivity imaging. Tree Physiol. 28, 1441–1448.

Rossi, R., Amato, M., Bitella, G., Bochicchio, R., Ferreira Gomes, J.J., Lovelli, S., Martorella, E., Favale, P., 2011. Electrical resistivity tomography as a non-destructive method for mapping root biomass in an orchard. Eur. J. Soil Sci. 62, 206–215. https://doi.org/10.1111/j.1365-2389.2010.01329.x

P2L19-20. The cited work from Cassiani does not report resistivity-root correlation/calibration. How could this paper demonstrate the reliability of the method as in Amato and coworkers?

*Cassiani et al. 2016, does not report resistivity root correlation/calibration. The analysis was conducted thanks to the monitoring of the soil water content due to RWU.*

*We rephrased the sentence.*

P2L23-26. is very interesting and you could also expand considering the case of soil with heterogeneous moisture or soil properties that would complicate further the calibration with root mass

*We do agree with this suggestion and added a sentence to the relevant paragraph.*

*"The problem would complicate further the correlation with root mass considering heterogeneous soil properties and moisture, and the electrical anisotropy caused by the roots system i.e. the root connectivity and root structure as further described in Rao et al. (2018)."*

P2L32. It is not clear if you are speaking of the resistivity of roots in soil (and at this point there is also the effect of the mass), or of a root fragment in itself.

The term dead wood is not correct, you could just use "heartwood and

the pith"

*The term dead has been removed from the text.*

*This paragraph indeed describes the literature of root fragment or wood material in itself.*

*The sentence was rephrased to avoid any confusion for the reader and make sure that the paragraph deals with root resistivity by itself.*

*"[…] an understanding of the contribution of the segments of the root system (by its own properties, with no interaction with soil) to that bulk signal is limited to only few studies describing wood electrical properties."*

P3L4-6. citations needed.

*We cite in the revised manuscript the work from York et al,. 2016. In this review, the temporal dynamics of rhizosphere activities (including root exudates) is considered, from annual fine root turnover to diurnal fluctuations of water and nutrient uptake. The latest empirical and computational methods are discussed in the context of rhizosphere integration.*

P3-L25. "sap flow processes take place at the cambium" is not correct,

sap flow hap- pens in xylem and phloem which are from either side of

the cambium.

*Sentence rephrased as follows: "In the plant stem and roots, electrical current is transmitted*

*through active electrical layers, in the xylem and phloem (on either side of the cambium), where sap flow processes take place."*

P3L27-28. "the quasi-infinite connection" is this your assumption? In this case needs to be clearly stated, or you could cite the paper that addresses this.

*Yes, this is our assumption. It has been clearly stated: "Our main assumption is to consider that thanks to the quasi-infinite fine root connections and their mycorrhizal at the interface between roots and soil, current tends to run out uniformly from the roots to the soil."*

*Note that this assumption was further strengthened following the recommendation of the reviewer 1 who suggested to add: "[…] Anderson and Higinbotham (1976) showed that maize roots have significantly lower electrical resistance in the radial than in the axial direction (thus being anisotropic), thus allowing current to exit laterally from the entire root length."*

P3L29-30. This may eventually be considered a strength of the method, as fine roots are the active ones, people in plant sciences would be interested to measure. You could put this more in evidence.

*This strength has been stated with a new sentence: "[…] This would be of major interest to measure for plant science community as fine roots are the active ones."*

P4L8. xylem and phloem sap, as xylem and phloem are not fluids. P4L27 Maybe this citation would be interesting for future readers:

Postic, F., Doussan, C., 2016. Benchmarking electrical methods for rapid estimation of root biomass. Plant Methods 12:33. https://doi.org/10.1186/s13007-016-0133-7

*One sentence was added to introduce the experimental work by Postic et al.: "A benchmarking of the experimental approaches supporting the subsequent theories is proposed in Postic et al., (2016)."*

Material and methods:

The biological material and agricultural system needs to be better described (how old are the plants, what is the distance between them, what shape size the canopy, what is the rootstock?)

*Sentence rephrased as follows: "*
*"Grapevine plants are planted with a distance of 1m between plants and 1.5m between rows. The vineyard is non irrigated. Considering also the selected plant and the slight slope of the vineyard, it might be reasonable to foresee a top layer rooting with an asymmetric development (gravitropism)*

2.1.1 Interpreted soil picture needed

P5L11. "colluviosol" is not international nomenclature. Please correct using IUSS Working Group WRB, 2014. World reference base for soil resources 2014. Interna- tional soil classification system for naming soils and creating legends

for soil maps, World Soil. ed. FAO, Rome

*We removed the "colluviosol" term.*

P5L11. Measurement of root density is key in this paper. How this was performed needs to be detailed.

*See our objection above about the value of direct measurements of field root distribution. Nevertheless, we have some observation of rooting depth from internal (confidential) documents. We added a sentence to provide more details on these observations.*

P5L13. What is "dense rooting for you?" please specify

*We removed the word "dense" and rephrased the sentence since the (confidential) report does not brings further quantitative information.*

P5L14. The term horizon instead of layer would be more scientific (from now on)
*Corrected*

P5L17. "terroir de grave" not clear what this is. Is this a term for a distinctive geological or pedological formation that is meaningful in local terminology? If this is a local term needs to be reported in italics and quoted

*Corrected (reported in italics and quoted). It is indeed a local term.*

P5L18. Only one plant??? That is far under the minimum for characterizing a biological systems, and even if you just want to show that the method is working how could you claim that is not just by chance?
*See previous answers*

P5L19. "sandy-clayey horizon" although correctly identifying and naming it would be better
*Corrected*

P5L21. English not clear (top layer rooting). Do you mean "asymmetric root develop- ment in the top layer"?
*Corrected*

Sections 2.2 and 2.3 In agreement with Reviewer 1 the geophysical acquisition and inversion procedure is not detailed enough to be replicated by new authors. This must be better detailed. Are you considering to share the MALM inversion code with this article?

*We followed also the recommendations of the reviewer 1 to improve section 2.2 and 2.3. See relevant replies.*

*The MALM inversion code is not ready to share as it is still under confidential development for the extension from ERT et EIT until published. We nevertheless clearly aim to share it in the right context.*

**Results:**

P10L9-11. A root system with a lateral extension of 0.5-0.9m appears reasonable (considering the density of plants in figure 2), however no active roots deeper

than 0.3m, in a non-irrigated permanent tree it is a very striking and conflicting result. In my opinion this is not true, as I am not even sure that in a tilled soil, with grass in the inter row (figure 2) you will find the majority of tree roots in the first 0.3m. The problem here is that authors did not excavate the rootzone to demonstrate their measurements. Very few examples are reported here:

*Opinions may be respectable, but are not science. Excavating is not the solution, as stated above. Yet, we might consider this in the future.*

*We would like to attract the reviewer's attention to some facts: the vines are very small trees; the climate is certainly pretty wet; the soil is predominantly clayey (even though not much at the specific location considered here). So… what about opinions? How confident is the reviewer that the roots (the ACTIVE roots – the reviewer does not seem to make difference between supporting roots – structural – and roots that perform RWU…) should be deeper than 0.3 m?*

- Using magnetic resonance imaging root activity is already observed at around 30cm depth in just 20 days old Lupinus alba seedlings, (Carminati A (2013) Rhizosphere wettability decreases with root age: a problem or a strategy to increase water uptake of young roots? Front. Plant Sci. 4:298. doi: 10.3389/fpls.2013.00298)

*Lupinus is a very different species in the sense that it is not a woody plant. The root system development in that case is really different and of course can easily reach larger depth (while the upper part stays smaller). Yet, we wish to draw the reviewer's attention to the critical effect of the environment, as roots are active systems. It is bizarre that he/she never acknowledges the effect of forcing constraints on the ultimate development of roots: 30 cm days in 20 days.. ok. Under which conditions????*

- Using modelling and isotopic approaches the mean relative contribution to transpiration of soil layers under 0.4m is estimated around 50% in Rothfuss and Javaux, 2017 Reviews and syntheses: Isotopic approaches to quantify root water uptake: a review and comparison of method Biogeosciences, 14, 2199–2224, 2017

*Same comment as above: the reviewer never acknowledges specific conditions… a component of scientific reasoning is missing.*

- Water depletion and contribution to plant water status is observed under grapevines at 1.5m using electrical resistivity tomography in Brillante, L., Bois, B., Lévêque, J., & Mathieu, O. (2016). Variations in soil-water use by grapevine according to plant water status and soil physical-chemical characteristics-A 3D spatio-temporal analysis.European Journal of Agronomy, 77, 122–135.

*Water depletion and contribution to plan water status can be observed at much larger depth that the rooting depth but the reviewer forgets that water is sucked from depth also as an effect of capillary forces. Roots can be shallow, and yet the hydraulic head gradient they create sucks water from depths much larger than rooting depth itself. This is common understanding in soil physics, is it not?*

- Old-school maps of grapevine roots down to a depth of 1.5m (in irrigated conditions) can be find in Rob M. Stevens, Tim Douglas Distribution of grapevine roots and salt under drip and full-ground cover microjet irrigation

systems. Irrig Sci (1994) 15:147- 152

In this study, Rob M. Stevens and Tim Douglas state that they found "*values of Lv and La* (read: root length densities) *which are higher than those previously reported for grapevines*". They also state that "*Under both irrigation systems, roots were concentrated near the surface with 74 and 72% of the vines' roots within 80 cm of the surface for microjet and drip, respectively.*" And from the graph we can see that the pick is located for 30/40 cm depth.

*Tomasi, D., Battista, F., Gaiotti, F., Mosetti, D., & Bragato, G. (2015). Soil influence on root distribution and implications for berry and wine quality of the Tocai Friulano variety. American Journal of Enology and Viticulture, ajev-2015.

*Again, the reviewer NEVER ACCOUNTS FOR SPECIFIC FORCING CONDITIONS, as if a species (vine) always developed in the same manner irrespective of climate/soil structure, etc.*

*PS: note that "Tocai Friulano" is an unlawful name since the early 2010 as a result of EU regulations in favor of the original Hungarian grape (Tocaj) ;-)*

P10L15-16. This work does not show that MALM can provide anything useful to dis- criminate spatial distribution of roots if not lateral extension within the first 30cm of soil depth.

*This is a rather arbitrary sentence, still based on the wrong assumption that excavating is the solution … Yet, we rephrased the sentence to read:*
*"Our work clearly shows that the MALM method can provide key information concerning the root system spatial distribution of tree woody species (with the latter discussed uncertainties)."*

P10L27-30. There is no evidence in this article "that injecting current in the plant stem causes a distribution of electrical current sources in the ground that correspond to the location of active roots." Although I would eventually agree with this author suggestion, there is not theoretical argumentation to sustain this in this text, neither experimental evidence.

*Theory or experiments? What is the reviewer seeking? He/she neglects our experimental evidence here? And invoke a theory nobody has written yet? Again, the same questionable view is underlying the reviewer's arguments.*

P11L5-14. This is well written

*Thanks*

P11L17-19. This is not true, as also through the bark there can be some, small water uptake, which could allow electrical flow. Cuneo et al., 2018, Water uptake can occur through woody portions of roots and facilitates localized embolism repair in grapevine. New Phytologist, 218(2)

*This statement has been moderated to read:*

*"[…] Water acquisition and by prolongation electrical current pathway is thought to be limited to the surface located close to root tips. At least two other phenomena may contribute to current release higher than expected. Firstly, Cuneo et al., (2018) show that although woody portions of roots act as an electrical barrier (also to microbial degradation), exchanges may occur during water uptake can occur through (in order to facilitates localized embolism repair in grapevine). Secondly, as discussed also in the introduction, some roots show anisotropic electrical conductivity, allowing current to flow radially more easily than longitudinally (Anderson and Higinbotham, 1976). In this case, our proposed MALM approach would need to be modified in the interpretation*

*stage. Note that roots are generally electrically anisotropic at the microscopic scale (few cm) and also macroscopically the root architecture and soil water uptake pattern can induce anisotropy. Using MALM to study anisotropy of root structures can indeed be a separate, very promising, area of research."*

P11L19-22. Interesting field of research

P11L24-25. Slightly narrower range?? See previous comment, also when referring to literature, please cite at least one article!

*We refer to the paper by Rob M. Stevens and Tim Douglas (2014) and Gerós et al., (2015) to support this sentence.*

P11L25-26. Out of context. No data to support the sentence.

Remove.

*Sentence removed*

P11L33. This is sure

P12L8. Interesting field of

research **Conclusion**:

Avoid to repeat the methods in the conclusion

*We have shortened the conclusions.*
https://doi.org/10.5194/hess-2018- 238, 2018.

*Please also note the revised manuscript red marked as a supplement to this comment*

[revised manuscript text omitted]

---

## Author Response (AR1)

**Small scale characterization of vine plant root water uptake via 3D electrical resistivity tomography and Mise-à-la-Masse method**

**Response to editor comments**

Dear Mr. Vanclooster,

We thank you and the reviewers for your kind reviews and constructive comments. We took notice of your suggested minor revisions. The lack of ground truth root system assessments based on excavation has been discussed more in detail in the discussion section of the paper (see L. 25 p.10 to 37 p.10 of the new manuscript).

Please find attached a revised version of the manuscript.

Best regards,

Benjamin Mary et al.

[revised manuscript text omitted]